# Regional climate imprints of recent historical changes in anthropogenic Near Term Climate Forcers

Alba Santos-Espeso[1,2], María Gonçalves Ageitos[1,2], Pablo Ortega[1], Carlos Pérez García-Pando[1,3], Markus G. Donat[1,3], Margarida Samso Cabré[1], and Saskia Loosveldt Tomas[1]

[1]Barcelona Supercomputing Center, Barcelona, Spain
[2]Universitat Politècnica de Catalunya, Barcelona, Spain
[3]ICREA, Catalan Institution for Research and Advanced Studies, Barcelona, Spain

**Correspondence:** Alba Santos-Espeso (alba.santos@bsc.es)

**Abstract.** Near-Term Climate Forcers (NTCFs) play a crucial role in shaping Earth's climate, yet their effects are often overshadowed by long-lived greenhouse gases (GHGs) when addressing climate variability. This study explores the climatic impact of elevated non-methane NTCF concentrations from 1950 to 2014 using CMIP6-AerChemMIP simulations. We analyse data from four Earth System Models with interactive tropospheric chemistry and aerosol schemes, leveraging a twelve-member ensemble to ensure statistical robustness. Unlike single-species or idealised radiative forcing studies, our approach captures the combined effects of co-emitted NTCF species. Our results show that the negative radiative forcing of aerosols dominates the overall NTCF impact, offsetting the warming effects of absorbing aerosols and tropospheric ozone. Multi-model mean analyses reveal three key regional climate responses: (1) a global cooling, amplified in the Arctic, where autumn temperatures decrease by up to 5°C, (2) a 38% increase in Labrador Sea ocean convection, and (3) changes in tropical precipitation, including a 0.6° southward displacement of the Intertropical Convergence Zone (ITCZ). This research addresses the mechanisms driving these climatic changes and underscores the importance of incorporating interactive NTCFs in climate projections. As inferred from their historical impact, future NTCF reductions could amplify regional responses to increasing GHG concentrations, thus requiring more ambitious mitigation strategies.

## 1 Introduction

Understanding the intricate dynamics of Earth's climate system and the influence of human activities is crucial for devising effective climate policies. A key message in the fight against global warming is the critical need to reduce anthropogenic atmospheric emissions. These emissions contribute to increased concentrations of species that directly or indirectly impact climate, broadly categorised into long-lived greenhouse gases (GHGs) and near term climate forcers (NTCFs). While long-lived GHGs, such as carbon dioxide ($CO_2$), are well-known for their persistent warming effects, NTCFs present a more complex and variable influence on the climate system.

NTCFs are chemically and physically reactive compounds whose impact on climate occurs primarily within the first decade after their emission (Myhre et al., 2013). They include methane ($CH_4$), tropospheric ozone, black carbon, organic carbon, sulphates, and other aerosols. Because of their short lifetimes, NTCFs, in particular aerosols, are heterogeneously distributed

in the atmosphere, having the potential to affect climate variability both globally and regionally. While $CH_4$ is both a potent

GHG and a NTCF due to its relatively short atmospheric lifetime compared to $CO_2$, it behaves differently from other NTCFs as it is well-mixed throughout the atmosphere. In this study, we focus on non-methane NTCFs.

The primary mechanism through which NTCFs influence climate is through modifications of the Earth's radiative balance. Tropospheric ozone acts as a GHG, while aerosols impact radiation through both direct and indirect effects. Directly, aerosols alter radiative forcing by either absorbing or scattering sunlight: black carbon absorbs radiation, contributing to warming,

whereas sulphates scatter radiation, leading to a cooling effect. Indirectly, aerosols influence cloud properties by acting as cloud condensation nuclei (CCN) and ice-nucleating particles (INP), enhancing cloud formation and altering cloud reflectivity, lifetime, and precipitation patterns. These aerosol-cloud interactions further modify radiative forcing, reinforcing the overall cooling effect of aerosols. (Szopa et al., 2021; Wall et al., 2022).

Through their interaction with radiation, changes in the spatio-temporal distribution and composition of NTCFs lead to

distinct climate responses. Several studies report NTCFs effects on atmospheric and oceanic circulations. Allen and Sherwood (2011), through sensitivity experiments with an atmospheric general circulation model, found nearly opposite responses in atmospheric circulation to radiation scattering or absorbing aerosols. While scattering aerosols reduce the Hadley cell width and displace the Intertropical Convergence Zone (ITCZ) southward, absorbing aerosols lead to a northward ITCZ shift. These responses are attributed in their study to interhemispheric temperature gradients arising from the spatially uneven distribution

of the radiative forcing. Similar ITCZ behaviour has been reported as a response to the asymmetric distribution of dust (Evans et al., 2020) and volcanic aerosols (Pausata et al., 2020) between hemispheres. On the other hand, a multi-model analysis suggests that black carbon and tropospheric ozone, both contributing to tropospheric warming, are the most likely causes of the observed poleward shift of the tropical circulation in the Northern Hemisphere from 1979 to 1999 (Allen et al., 2012).

In the ocean, aerosols affect the Labrador Sea convection and the Atlantic Meridional Overturning Circulation (AMOC).

Studies based on CMIP5 and CMIP6 multi-model analyses suggest that increasing trends in global aerosol concentrations strengthened the AMOC between 1850 and 1985 (Menary et al., 2020; Robson et al., 2022). And in the same line, future NTCFs reductions may enhance the projected AMOC weakening (Hassan et al., 2022). Recent research by Liu et al. (2024) found that Asian aerosol forcing has opposite effects on AMOC compared to those of emissions from Europe and North America. Their study, examining the AMOC slowdown from the mid-1990s as well as future projections, indicate that Asian aerosols hinder

Labrador Sea convection, contributing to an AMOC slowdown. This result is particularly significant as Asia, despite recent declines, has become a primary region of anthropogenic aerosol emission, whereas until the 1990s, emissions were dominated by non-Asian sources. Building on this regional distinction, Cowan and Cai (2013) used a coupled atmosphere-ocean model and showed that non-Asian aerosols dominated the ocean response to global aerosol forcing during the 20th century, delaying the GHG-induced weakening of the meridional overturning circulation and, consistently, increasing the northward heat transport

across the equatorial Atlantic.

Another known hotspot for NTCF impacts is the Arctic. Black carbon and tropospheric ozone emissions contribute to Arctic surface warming, opposing the cooling effect of global tropospheric aerosols (Quinn et al., 2008; Sand et al., 2016). Krishnan et al. (2020) examined the mechanisms through which recent European aerosol reductions may have caused Arctic warming,

giving great relevance to poleward heat transport changes. Using slab-ocean simulations to isolate atmospheric and ocean contributions, they found that Arctic warming is primarily driven by atmospheric turbulent fluxes and their interaction with sea ice, while ocean heat convergence produces a cooling effect. In contrast, Acosta Navarro et al. (2016) found that enhanced oceanic heat transport played a greater role, increasing Arctic energy intake and triggering sea ice responses. Regardless, of the source of the anomalies, the Arctic magnifies temperature changes through different active positive feedbacks, a phenomenon known as Arctic Amplification (Previdi et al., 2021). In the Arctic, temperature changes are predominantly confined to the lower troposphere due to strong surface-based processes and seasonal stratification, particularly during boreal autumn and winter. The lapse rate feedback in the Arctic is characterised by stronger temperature changes in the lower levels as compared to upper troposphere. The vertical temperature gradient modulates outgoing long-wave radiation, amplifying temperature variations (Boeke et al., 2021). Closely linked to this mechanism is the albedo feedback, where changes in sea ice extent regulate local energy intake during the light seasons, due to its higher albedo compared to the ocean surface. This enhances temperature variations, especially during darker seasons, when the ocean-atmosphere energy transfer occurs (Feldl et al., 2020).

To better understand and account for these complex interactions, Earth System Models (ESMs) are essential tools for studying NTCF impacts on climate. By simulating the interplay between atmospheric, oceanic, and terrestrial components, these models provide valuable insights into climate sensitivities and feedback mechanisms. Collaborative initiatives such as the Coupled Model Intercomparison Project Phase 6 (CMIP6; Eyring et al., 2016) play a crucial role in advancing climate research by standardizing experimental frameworks, refining future scenarios, and enabling systematic model intercomparisons.

Within CMIP6, historical simulations are a flagship set of experiments designed to evaluate ESM performance against observations and to investigate the role of external forcings in shaping the climate of the industrial era (1850-2014). These simulations incorporate estimates of past changes in relevant forcers, capturing human-induced changes in GHG concentrations and NTCFs. The Aerosols and Chemistry Model Intercomparison Project (AerChemMIP; Collins et al., 2017; Griffiths et al., 2025), endorsed by CMIP6, specifically targets NTCFs to quantify the climate and air quality impacts of aerosols and chemically reactive gases through a range of dedicated simulations.

Through the analysis of CMIP6-AerChemMIP simulation data, this study provides a comprehensive and quantitative assessment of NTCF impacts on the global climate system. Our multi-model analysis focuses on three main climate responses: pronounced Arctic cooling, increased Labrador Sea convection, and a southward displacement of the ITCZ. Section 2 describes our approach, Section 3 presents key findings, and Section 4 discusses their implications and potential directions for future research.

## 2   Methodology

In the following subsections we describe the selection of model data, the statistical metrics applied, and key diagnostics used to assess NTCF impacts on specific aspects of climate such as ocean density and the ITCZ. All analyses were conducted using the Earth System Model Evaluation Tool (ESMValTool; Righi et al., 2020), an open-source tool that ensures consistent, traceable, and reproducible processing of multi-model climate data.

## 2.1 Model selection and experimental design

For this study, we selected ESMs with interactive tropospheric chemistry and aerosols that contributed to two different CMIP6 experiments: *historical* and *hist-piNTCF* (Eyring et al., 2016; Collins et al., 2017). *hist-piNTCF* uses the same historical forcings as *historical* except for anthropogenic non-methane NTCFs emissions (aerosols, tropospheric ozone and their precursors), which are instead fixed at 1850 values. Therefore, *hist-piNTCF* omits the increased NTCF concentrations that are present in the *historical* experiment while maintaining other radiative forcers such as the well-mixed GHG (Hoesly et al., 2018). By comparing these two types of simulations, we can isolate the effects of NTCFs on historical climate variability.

Using multiple ESMs enables us to assess the robustness and inter-model consistency of the NTCF signal while accounting for uncertainties from structural model biases. Although additional AerChemMIP experiments (e.g., *hist-piAer*) could help disentangle the effects of individual species, their limited availability across models would require reducing the ensemble size or breaking the consistency between experiment sets, compromising comparability. Focusing on NTCFs as a whole therefore provides a balanced perspective: it allows us to assess the combined effect of short-lived warming and cooling species–particularly relevant from a policy perspective–while leveraging a reasonably large ensemble of models. This approach complements and builds upon existing attribution studies that isolate the effects of individual forcers (e.g., Allen and Sherwood, 2011; Menary et al., 2020; Szopa et al., 2021; Zhang et al., 2021; Wu et al., 2024).

We selected ESMs that provided at least three members for each experiment, i.e. the minimum requested by the AerChemMIP exercise. This requirement allows us to better constrain the forced signals by averaging out some of the internal variability that emerges spontaneously in each member due to the stochastic nature of the climate system (Tebaldi and Knutti, 2007). Considering all the previous points, the models included into the study are BCC-ESM1 (Wu et al., 2020), MRI-ESM2-0 (Yukimoto et al., 2019), UKESM1-0-LL (Sellar et al., 2019), and EC-Earth3-AerChem (van Noije et al., 2021).

These models represent a diverse set of contributions from different institutions, with varying ocean and atmospheric physical components, as well as atmospheric chemistry schemes (see Table A1). All four models include comprehensive gas-phase chemistry schemes that allow deriving tropospheric ozone concentrations. They also resolve key aerosol species, both anthropogenic and natural (i.e., dust and sea salt) taking into account their interactions with clouds and radiation. By representing these real life processes the models are able to capture possible feedbacks and indirect impacts of the applied forcings, which makes them suitable for the purpose of this study (Huijnen et al., 2010; Yukimoto et al., 2019; Mulcahy et al., 2020; Wu et al., 2020).

The analysis focuses on the period from 1950 to 2014, during which the availability of satellite and higher-quality observational data improved confidence in the forcing estimates used in climate models, thereby enhancing the reliability of our results (Yang et al., 2016). Over this timeframe, atmospheric composition varied significantly. Global aerosol concentrations increased in the early decades, followed by a stabilisation from 1980s onward, with regional differences in anthropogenic emissions. While Europe and North America implemented aerosol reduction measures, Asian emissions continued to rise (Tørseth et al., 2012; Klimont et al., 2017; Aas et al., 2019), although, recent studies indicate that CMIP6 forcing datasets underestimate China's reductions in anthropogenic aerosol emissions during 2006-2014 (Wang et al., 2021). This potential bias should be

considered when interpreting our results. In contrast, GHG concentrations, including tropospheric ozone, showed a continuous increase throughout the study period (Bauer et al., 2020; Griffiths et al., 2021). These divergent trends are particularly relevant, as most aerosol species and GHGs exert opposing radiative effects, making their combined influence on climate a key aspect of our analysis. For completeness, we note that the emissions for each experiment are prescribed following the CMIP6 AerChem-

130 MIP protocol (Collins et al., 2017), and the time evolution of different NTCF species emissions is documented in Hoesly et al. (2018). Furthermore, a regional decomposition of the emissions can be found in Fig. 6.19 of Szopa et al. (2021).

## 2.2 Statistical analysis

To assess the influence of NTCFs on key climate variables (e.g., temperature, precipitation, sea ice concentration), we analyse three main aspects: climatological mean differences, temporal variance changes, and annual value differences (Table 1). For

each analysis, we first compute annual or seasonal means depending on the climate feature of interest. Atmospheric variables are interpolated onto a $2° \times 2°$ grid to facilitate inter-model comparison, while oceanic and sea ice variables are retained at their native resolution to preserve the integrity of their spatial discretization.

**Table 1.** Statistical methods used to quantify and evaluate the confidence of NTCF-induced changes in climate.

| Signal | Confidence evaluation |
|---|---|
| Climatological mean difference | two-sample t-test (95% significance) |
| Temporal variance ratio | ensemble agreement (80-100% confidence) |
| Annual difference | two-sample bootstrap test (95% significance) |

For climatological means, we evaluate the direct difference between the two experiment ensembles (Eq. (1)):

$$\Delta X = X_{historical} - X_{hist-piNTCF} \tag{1}$$

where *X* represents any given climate variable. The statistical confidence of the mean signal over the studied period is assessed using a paired-samples t-test at 95% significance level. This test accounts for potential model-dependent differences in mean states by pairing the samples and evaluating changes between experiments model by model.

Changes in temporal variance can indicate alterations in physical processes or destabilisation of climate systems. To investigate this aspect, we study the standard deviations in time and compute the ratio between experiments (*R*). To facilitate

interpretation, we express the variability changes due to NTCFs as a percentage (Eq. (2)):

$$R = \frac{\sigma_{historical}}{\sigma_{hist-piNTCF}} \qquad \rightarrow \qquad \text{Variability change (\%)} = (R-1) * 100 \tag{2}$$

A positive percentage indicates increased variability due to NTCFs, while a negative percentage denotes reduced variability. Ensemble consistency is assessed based on the number of members agreeing on the sign of the response. Since the ensemble

consists of twelve members, agreement in 10 out of 12 members indicates a $\sim$ 80% confidence, 11 out of 12 members a $\sim$ 90% confidence, and full agreement up to $\sim$ 100% confidence.

For annual differences between experiments, we evaluate the means of 3 members per model and 12 members for the multi-model ensemble, which amount to relatively small sample sizes. To address this limitation, we employ a two-sample bootstrap test with 5000 resamples and a 95% significance level (Efron, 1979; Mudelsee and Alkio, 2007). This method generates new combinations of the *historical* and *hist-piNTCF* samples, preserving original sample sizes. For the multi-model mean, while the model contributions may vary across iterations we ensure it remains equal between samples. A difference between experiments is deemed statistically significant if zero falls outside the 95% confidence interval of the compiled 5000 resampled differences.

## 2.3 Thermal and haline contributions to ocean density

To evaluate the impact of NTCFs on ocean stratification, particularly in the Labrador Sea, we compute ocean density from potential temperature (*thetao*) and salinity (*so*) data using the polyTEOS10_bsq equation, a 55-term polynomial expression for density (Roquet et al., 2015). Additionally, to determine whether changes in stratification are driven by temperature or salinity variations, we calculate *sigmaT* and *sigmaS*, which represent the respective contributions of temperature and salinity changes to density (Bilbao et al., 2021). These values are derived using the thermal expansion (*a*) and haline contraction (*b*) coefficients, both computed as polynomial coefficients within the polyTEOS10_bsq framework (Eqs. (3) and (4)):

$$a = -\frac{\partial r}{\partial C_T} \quad [kg \cdot m^{-3} \cdot K^{-1}] \tag{3}$$

$$b = \frac{\partial r}{\partial S_A} \quad [kg \cdot m^{-3} \cdot psu^{-1}] \tag{4}$$

where $r$ is the density anomaly, $C_T$ is the conservative temperature and $S_A$ is the absolute salinity.

In particular, we compute the potential density anomaly with reference pressure of 0 dbar (*sigma0*). To facilitate interpretation, we normalise the data so that the normalised density anomaly (*sigma*) is expressed as the direct sum of *sigmaT* and *sigmaS* (Eqs. (5), (6) and (7)):

$$sigma = sigma0 - DenRef \tag{5}$$

$$sigmaT = -a \cdot (thetao - TempRef) \tag{6}$$

$$sigmaS = b \cdot (so - SalRef) \tag{7}$$

where *DenRef*, *TempRef* and *SalRef* represent the vertical mean climatological values of ocean density, temperature, and salinity, respectively.

## 2.4 ITCZ characterisation

A key objective of this study is to assess changes in equatorial precipitation resulting from the presence of NTCFs. Following methodologies similar to Frierson and Hwang (2012) and Donohoe et al. (2019), we analyse the full precipitation distribution

rather than focusing solely on the latitude of maximum precipitation, as is commonly done. This approach allows us to capture not only latitudinal displacements of the ITCZ but also any potential impacts on the equatorial rainfall amount.

To evaluate tropical precipitation changes, we employ two ITCZ-related indices. First, we calculate the zonal mean precipitation from 20° S to 20° N. Then, we determine the coordinates of the precipitation centroid (PCENT), defined as the point that delineates regions of equal weight in the precipitation distribution. By comparing PCENT coordinates between the *historical* and *hist-piNTCF* ensembles, we quantify the effects of historical NTCFs on the ITCZ latitude ($\Delta lat$) and equatorial precipitation amount ($\Delta pr$). The indexes are defined as follows (Eqs. (8) and (9)):

$$\Delta lat = lat(PCENT_{historical}) - lat(PCENT_{hist-piNTCF}) \tag{8}$$

$$\Delta pr = pr(PCENT_{historical}) - pr(PCENT_{hist-piNTCF}) \tag{9}$$

This refined approach provides a more comprehensive assessment of ITCZ shifts and their implications for tropical precipitation patterns.

## 3 Results and Discussion

### 3.1 Global signals

To assess climatic responses to NTCFs, we compare the multi-model ensemble means of the *historical* and *hist-piNTCF* experiments. We analyse the mean state differences and the variance changes in key variables, namely surface air temperature (*tas*) and precipitation (*pr*), which together provide an overall view of the main physical responses.

Our results reveal three prominent climatic signals (Fig. 1). Firstly, higher concentrations of NTCFs induce a global cooling effect, most pronounced in the Arctic (Fig. 1b). We attribute this enhanced regional cooling to aerosols (Lewinschal et al., 2019; Westervelt et al., 2020; Szopa et al., 2021), counteracting the warming effects of tropospheric ozone in this region (Sand et al., 2016). The Arctic response is likely magnified by Polar Amplification mechanisms (Previdi et al., 2021), further examined in Subsection 3.2.

Secondly, we detect an increase in *tas* variability over the Labrador and Norwegian Seas, key regions of deep water formation (Fig. 1d). This variance increase concentrates on multidecadal scales (not shown), consistent with the characteristic timescales of North Atlantic ocean circulation and convection. In fact, similar connections between aerosol forcing and enhanced ocean convection have been identified in previous studies (Delworth and Dixon, 2006; Iwi et al., 2012). The detected NTCF signal is explored further in Subsection 3.3.

Lastly, in the Tropics, historical NTCFs induce a notable decrease in precipitation north of the equator and an increase to the south, with no clear changes in precipitation variance detected (Fig. 1f,h). This pattern is consistent with a southward displacement of the ITCZ, a phenomenon observed in response to aerosol increases in previous studies (Pausata et al., 2020; Zhao and Suzuki, 2021), despite potential opposing influences from tropospheric ozone (Allen et al., 2012). This response is discussed in detail in Subsection 3.4.

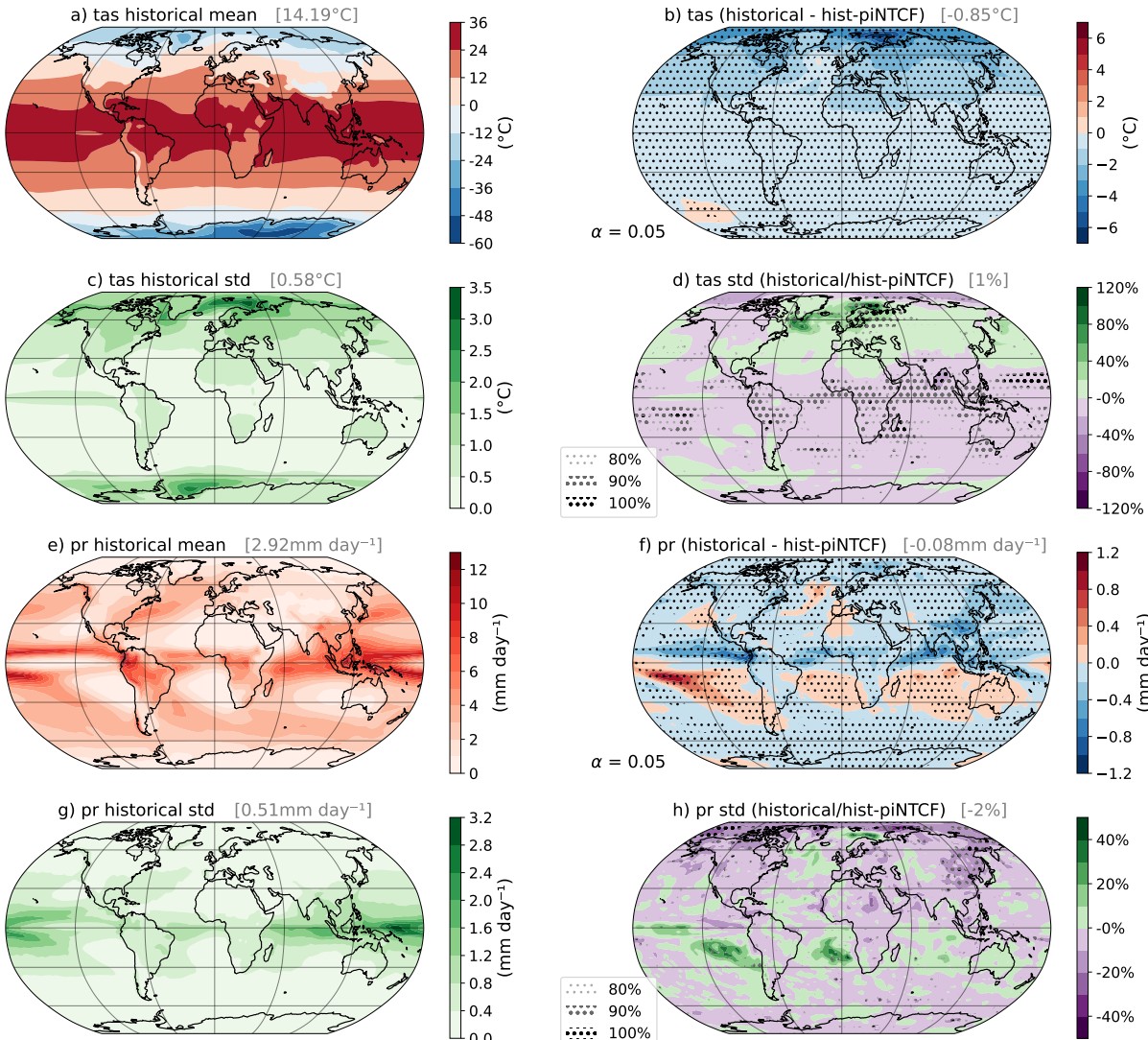

**Figure 1.** Impact of historical NTCFs on annual surface air temperature (*tas*; a, b, c, d) and precipitation (*pr*; e, f, g, h) as derived from the comparison of *historical* and *hist-piNTCF* CMIP6 simulations over the period 1950-2014. (a, e) Climatology for the multi-model *historical* mean and (b, f) difference in climatologies between the multi-model *historical* and *hist-piNTCF* ensemble means. (c, g) Standard deviation in time for the multi-model *historical* ensemble mean and (d, h) temporal variance ratio between the *historical* ensemble mean and its *hist-piNTCF* counterpart (expressed as percentage change). Global mean values for each magnitude are shown in gray in the title. The *historical* and *hist-piNTCF* ensembles analysed are comprised of 4 models (BCC-ESM1, MRI-ESM2-0, UKESM1-0-LL and EC-Earth3-AerChem) with 3 members each. Stippling is applied to significant values according to a paired sample t-test with a 95% confidence (b, f) and different percentages of ensemble members coinciding in the sign of the response (d, h).

## 3.2 NTCFs impact on Arctic temperature

Delving into the Arctic *tas* signal (Fig. 1b), we observe that the cooling in the *historical* ensemble, compared to *hist-piNTCF*, is most pronounced at the lowest levels of the atmosphere between 70°N and 90°N (Fig. 2). Regarding the season, the cooling peaks in boreal autumn (up to -5°C difference), while in summer, the strongest anomalies shift towards lower latitudes (Fig. B1). This temperature behaviour aligns with Arctic Amplification (AA), with the strongest temperature changes occurring near the surface, and seasonal feedbacks causing greater amplification in autumn and winter (Previdi et al., 2021). This cooling is consistent with the expected influence of higher aerosols concentrations in the *historical* ensemble, further amplified through sea ice–associated feedbacks.

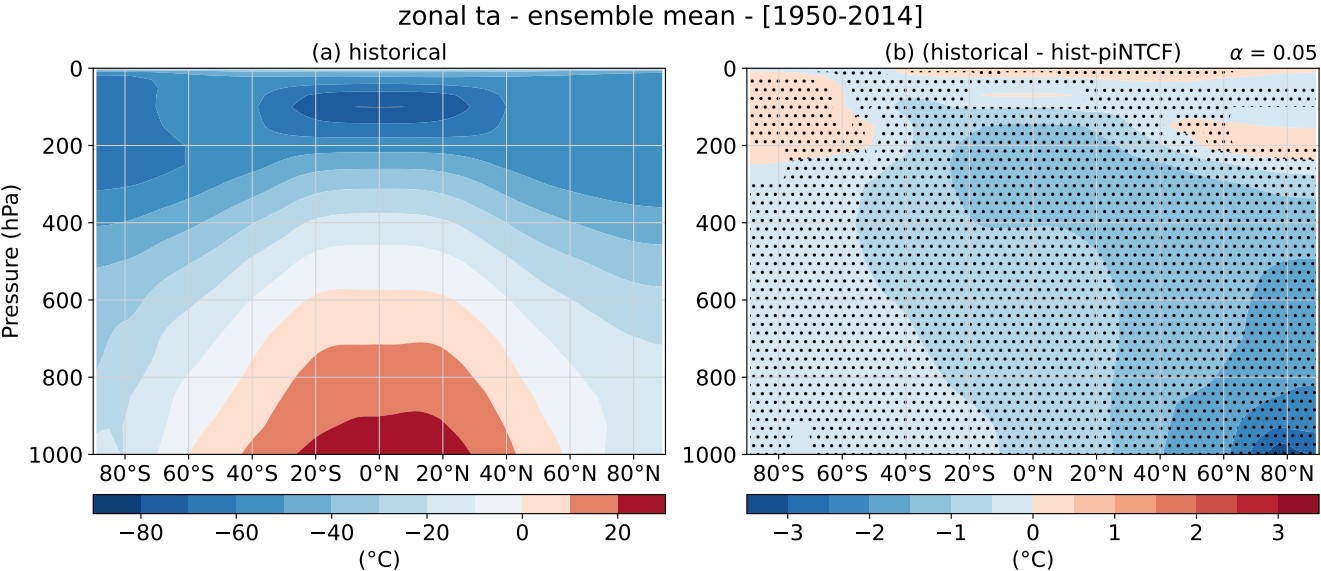

**Figure 2.** Impact of historical NTCFs on annual zonal mean air temperature (*ta*) over the period 1950-2014. (a) Climatology for the multi-model *historical* ensemble mean and (b) difference between the *historical* and *hist-piNTCF* ensemble means. The ensembles analysed are comprised of 4 models (BCC-ESM1, MRI-ESM2-0, UKESM1-0-LL and EC-Earth3-AerChem) with 3 members each. Stippling is applied to significant values according to a paired sample t-test with a 95% confidence.

The temporal evolution of the global and Arctic temperature responses (Fig. 3) reveals two distinct phases: from 1950 to the 1980s, the *historical* ensemble shows a cooling trend, whereas the *hist-piNTCF* ensemble experiences a slow temperature increase; after the 1980s, both ensembles show similar warming trends. This trend change is particularly evident in the differences between ensembles (Fig. 3b,d), which closely follow historical aerosol concentration trends (Fig. B2b). In fact, a strong anti-correlation (r=-0.86) highlights the coupling between the *od550aer* and *tas* global signals across the ensemble members available for both variables (see Fig. B2 caption). Zhang et al. (2021) explicitly attribute the "pothole-shaped" temperature evolution seen in historical experiments between 1960 and 1990 (Fig.. 3a,c) to aerosols, arguing that excessive sulphate load-

ing caused CMIP6 models to overestimate the aerosol-induced cooling anomaly, which is absent in aerosol-free experiments. The continued increase in ozone and other greenhouse gases (GHGs) provides a plausible explanation for the offset of the NTCF-induced cooling trend after 1980.

To quantify the AA attributable to NTCFs, we compute the Arctic Amplification Factor (AFF; Wu et al., 2024) (Eq. (10)):

$$AAF = \frac{m(\Delta T_{arctic})}{m(\Delta T_{global})} \tag{10}$$

where $\Delta T_i$ represents the temperature difference between the *historical* and *hist-piNTCF* ensembles in the different regions (Fig. 3b,d), and $m$ represents the slope of these signals (linear trends). For the period 1950–1980, the AAF of NTCFs is 3.87 for the multi-model mean (see Table A2 for individual model values), indicating that Arctic cooling due to NTCFs was nearly four times stronger than the global average. This aligns with a previous quantification of 3.87±0.48 for anthropogenic aerosol forcing during a comparable time period (Wu et al., 2024), highlighting the dominant role of aerosol forcing amongst the different NTCFs species. After the 1980s, however, this forcing diminishes in significance, as GHG driven warming becomes the primary driver of both Arctic and global temperature trends.

A recent study (Wu et al., 2024) has found that AA due to anthropogenic aerosols exceeds that induced by GHGs because of stronger feedback sensitivity to aerosol cooling. In particular, the study suggests sea ice-related feedbacks to be more effective in response to aerosols. Indeed, the NTCF-induced cooling in Fig. 1b aligns with an increase in sea ice concentration (*siconc*). Examining boreal autumn data—when sea ice retreat peaks (Deser et al., 2010)—reveals a consistent increase in sea ice extent in the *historical* ensemble relative to *hist-piNTCF* across multiple models (Fig. B3). The strongest increase occurs in the Barents Sea, a region known for its high sensitivity to external forcing and large sea ice internal variability (Rieke et al., 2023; Siew et al., 2024), and spatially aligns with the most pronounced temperature decreases (Fig. 1b). While this co-variability is consistent with the operation of local positive sea ice-related feedback mechanisms, the observed changes likely reflect the combined influence of several processes governing the cooling and its amplification (Previdi et al., 2021).

Additionally, Arctic sea ice expansion may contribute to the tropical cooling signal observed in Fig. 2. This vertical structure, showing amplified temperature anomalies in the tropical upper troposphere, is consistent with the operation of moist-adiabatic lapse rate adjustments (Colman and Soden, 2021). The pattern also aligns with the response to Arctic sea ice loss reported by England et al. (2020). They link Arctic sea ice loss to a slowdown in subtropical meridional ocean circulation, reducing equatorial upwelling and warming the tropical atmosphere, suggesting that the enhanced Arctic sea ice extent in the *historical* ensemble (Fig. B3) may further contribute to the observed cooling.

Overall, our results show that NTCFs produced Arctic cooling with strong amplification between 1950 and 1980. The observed increase in sea ice extent spatially aligns with the temperature response, which could suggest the operation of sea ice-related feedback mechanisms. Other processes that have been previously invoked to explain Arctic Amplification, such as changes in atmospheric and oceanic poleward energy transports (Iwi et al., 2012; Robson et al., 2022; Needham and Randall, 2023), cloud and water vapour feedback (Goosse et al., 2018), and lapse-rate feedback (Pithan and Mauritsen, 2014) may have contributed to the pronounced regional temperature changes. Quantifying their relative influence, however, lies beyond the scope of this analysis.

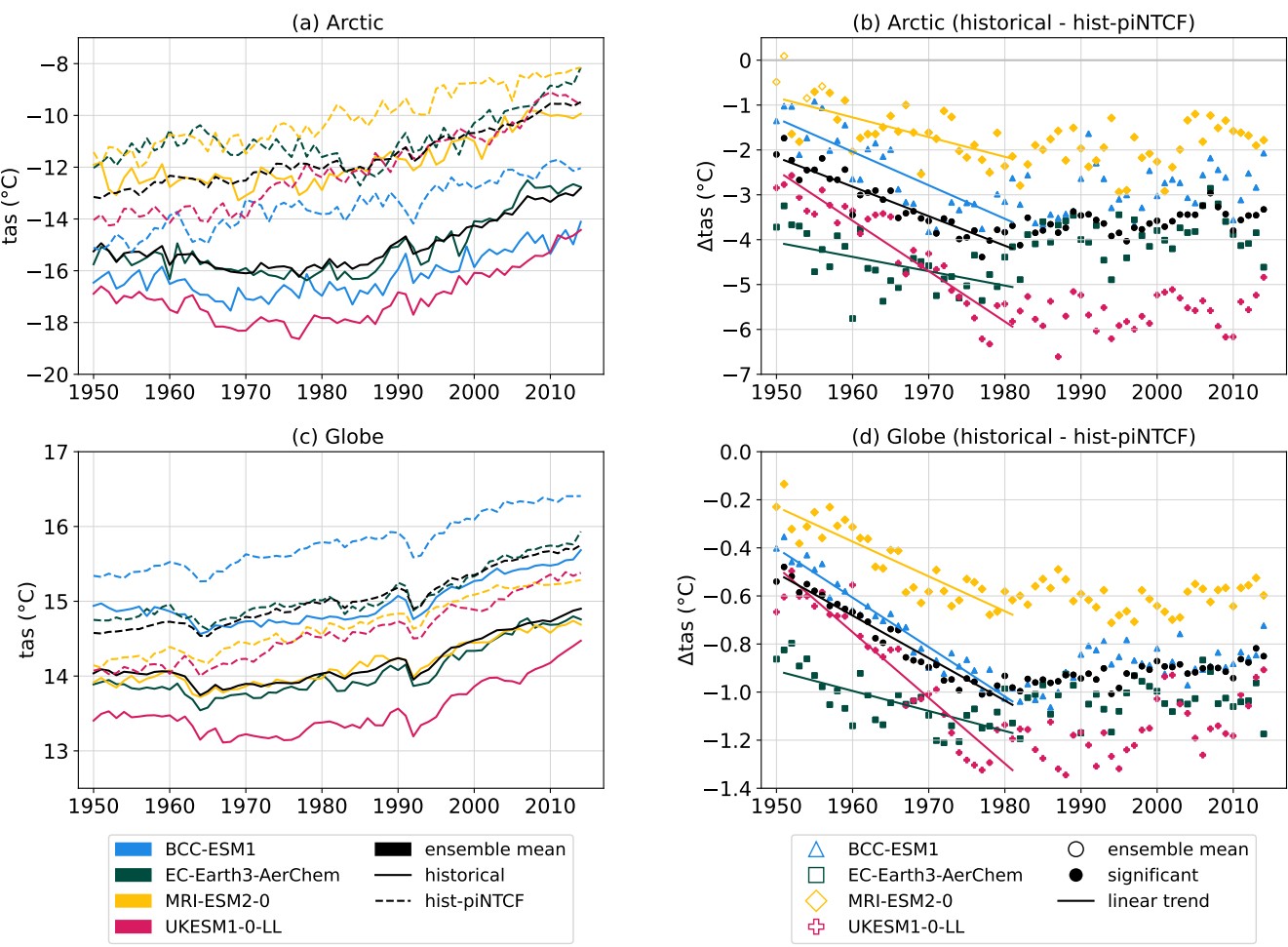

**Figure 3.** Impact of NTCFs on the Arctic (70°N–90°N; a,b) and global (c,d) surface air temperature (*tas*), as derived from CMIP6 simulations over the period 1950–2014. (a,c) Annual means from *historical* (solid) and *hist-piNTCF* (dashed) CMIP6 experiments (b,d) and their difference (symbols). In panels (b,d) solid lines represent the linear trends for the period 1950–1980 and filled symbols indicate significant values based on a bootstrapping significance test with 95% confidence (see subsection 2.2). Colours and shapes represent data from individual models (BCC-ESM1: blue triangles, EC-Earth3-AerChem: green squares, MRI-ESM2-0: yellow diamonds, UKESM1-0-LL: pink crosses) and black circles show the multi-model mean. For each experiment and model we consider the mean of 3 members.

### 3.3 NTCFs impact on Labrador Sea convection

Our results suggest that historical NTCFs enhanced surface temperature variability in key regions of deep water formation of
the subpolar North Atlantic (SPNA; Fig. 1d). To better understand this surface signal, we assess changes in the mixed layer depth (*mlotst*), a widely used proxy for oceanic convection. The analysis shows that higher historical NTCF concentrations led

to increased convection in the Labrador Sea across all models considered (Fig. 4). Additionally, a pronounced deepening of convection is observed in the Greenland Sea in all models except BCC-ESM1. The months of February, March and April are the focus of this analysis, as they correspond to the peak convection season in the Labrador Sea (Fig. B4).

Notably, EC-Earth3-AerChem displays a unique behaviour, with two out of three *historical* ensemble members showing episodes of collapsed convection in the Labrador Sea (Fig. B5), a phenomenon absent in the *hist-piNTCF* members. This behaviour is consistent with known Labrador Sea convection shutdowns in EC-Earth3-models that can persist for extended periods (Bilbao et al., 2021; Döscher et al., 2022). Meccia et al. (2023) attributes these episodes to a multi-centennial oscillation triggered by the accumulation of salinity anomalies in the Arctic that, when released into the North Atlantic, affect water

column stability and therefore convection. Importantly, they find that future scenarios with warmer climates lack sufficient sea ice to trigger the collapsing mechanism, potentially explaining its absence on hist-piNTCF members. Due to the strong dependency of the collapse episodes on internal variability, the specific response of convection to anthropogenic NTCFs is not correctly reflected in Fig. 4d and is likely underestimated. Consequently, the following analyses consider separately the EC-Earth3-AerChem member that maintains active convection (denoted by thin lines).

The temporal evolution of *mlotst* in the Labrador Sea (Fig. 5) provides further insights. Despite differences in their mean states, all models show comparable and significant responses to NTCFs. The *hist-piNTCF* experiments show a decrease in convection, in line with the expected response to rising GHG concentrations. In contrast, all *historical* experiments show stable or increasing *mlotst* values except for MRI-ESM2-0 (Fig. 5a). This model reports increasing convection until the 1980s after which convection declines, aligning with a first period of increasing global aerosol concentrations followed by a second

period with stabilised aerosol concentrations and stronger GHG forcing (Fig. B2). This suggests NTCFs counteracted, or at least mitigated, the GHG-driven decline in convection. The difference signal (Fig. 5b) shows a persistent enhancement of convection with decadal oscillations that are not in phase across models. To quantify the convection increase in response to NTCFs, we define the Labrador Sea Convection Response (LSCR) using a linear approximation (Eq. (11)):

$$LSCR\,(\%) = \frac{mN}{LSC_{clim}} \times 100 \qquad\qquad\qquad (11)$$

where $LSC_{clim}$ represents the mean Labrador Sea *mlotst* during the first decade in *hist-piNTCF*, $m$ denotes the slope of the *historical* minus *hist-piNTCF* difference (Fig. 5b), and N equals 65 years. The multi-model mean suggests a 38% increase in Labrador Sea convection due to NTCFs from 1950 to 2014 (individual model LSCR values are provided in Table A3).

     To better understand the reasons for the consistent model response in mixed layer depth, we study the vertical profiles of potential temperature (*thetao*), salinity (*so*), and potential density (*sigma0*; see subsection 2.3) in the Labrador Sea (Fig. 6).

Compared to *hist-piNTCF*, the *historical* ensemble exhibits colder and saltier near-surface conditions. Both contributing to higher surface density, these factors are linked to weaker local stratification and therefore intensified convection (as observed in Fig. 4).

     The saltier surface conditions may result from a positive feedback: stronger convection, initially driven by surface cooling, brings saltier subsurface waters to the surface, further increasing surface density and reinforcing deep convection. Although our

analysis based on monthly model outputs does not allow us to clearly separate the driving signals of deep convection from the

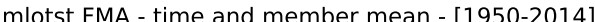

**Figure 4.** Impact of historical NTCFs on ocean mixed layer thickness defined by sigma T (*mlotst*) in the months of February, March and April (FMA), as derived from the comparison of *historical* and *hist-piNTCF* CMIP6 simulations over the period 1950–2014. (a, c, e, g) FMA climatology for the *historical* experiment and (b, d, f, h) difference in climatologies between the *historical* and *hist-piNTCF* ensembles. For each experiment and model we consider the mean of 3 members. Stippling is applied to significant values according to a two independent samples t-test with a 95% confidence (b, d, f, h). The black box limits the Labrador Sea area (60°W, 45°W; 50°N, 65°N).

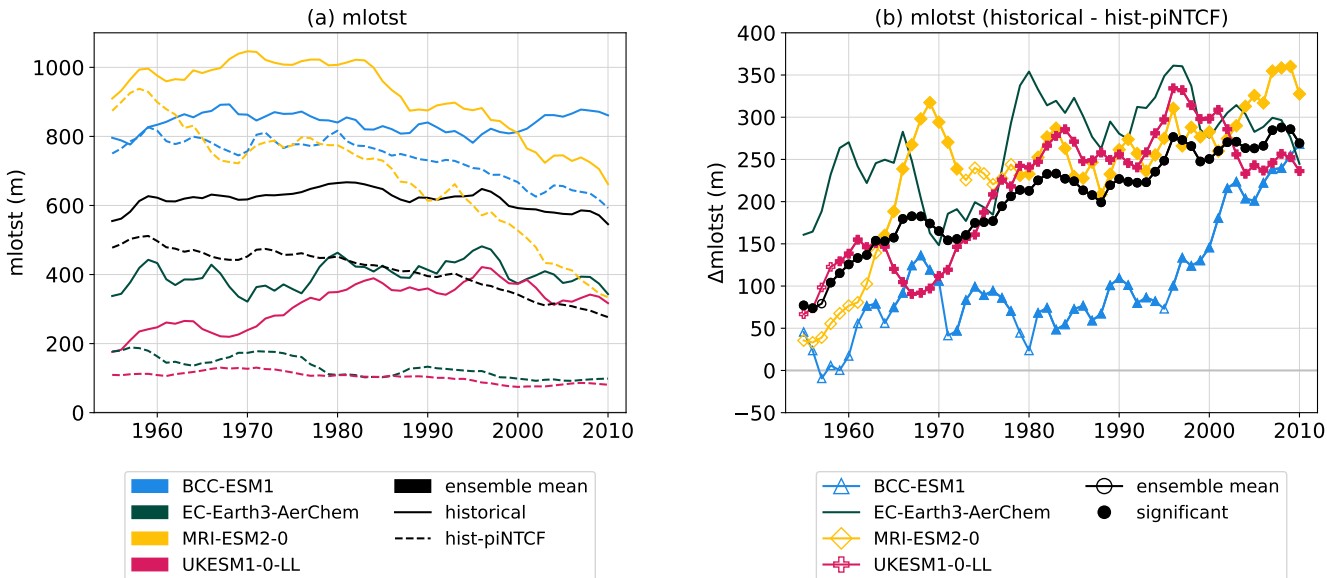

**Figure 5.** Impact of NTCFs on the Labrador Sea (60°W, 45°W; 50°N, 65°N) convection during February, March and April (FMA), as derived from the comparison of *historical* and *hist-piNTCF* CMIP6 simulations over the period 1950–2014. (a) Ten-year rolling mean of the ocean mixed layer thickness (*mlotst*) for *historical* (solid) and *hist-piNTCF* (dashed) experiments and (b) their difference, where filled symbols indicate significant values based on a bootstrapping significance test with 95% confidence (see subsection 2.2). Colours and shapes represent data from individual models (BCC-ESM1: blue triangles, EC-Earth3-AerChem: green, MRI-ESM2-0: yellow diamonds, UKESM1-0-LL: pink crosses) and black shows the model mean. For each experiment and model we consider the mean of 3 members, except EC-Earth3-AerChem with only one member (thin lines, no significance applied).

resulting response, a similar feedback mechanism has been identified in idealised frameworks (Lenderink and Haarsma, 1994), suggesting that this process is plausible in regions such as the Labrador Sea where subsurface waters are climatologically saltier (Fig. 6b). This mechanism could also explain the steady increase in *mlotst* seen in Fig. 5b, despite aerosol reductions after the 1980s.

The monthly evolution of potential density and its temperature and salinity contributions (Fig. B6; see subsection 2.3) provides additional insight into the processes driving convection, despite methodological limitations. We observe that NTCFs enhance the temperature contribution to surface density increase during the non-convective months (red profiles in Fig. B6e). During this summer period, warmer surface temperatures maintain water-column stratification, therefore, greater surface cooling due to NTCFs in the *historical* ensemble would increase surface density and erode the stratification, favouring convection

in the subsequent months. As convection activates, the relative contribution of salinity increases, consistent with the seasonal salinity changes arising from sea ice formation and melting, potentially relevant as the historical presence of NTCFs results in greater sea ice extent in the Labrador Sea region (Fig. B3). Further analysis would be required to quantify the sea ice

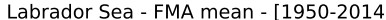

**Figure 6.** Impact of historical NTCF forcings on ocean profiles in the Labrador Sea Area (60°W, 45°W; 50°N, 65°N) during February, March and April (FMA), as derived from the comparison of *historical* and *hist-piNTCF* CMIP6 simulations over the period 1950–2014. (a, b, c) Climatology for the *historical* experiment, and (d, e, f) mean difference between *historical* and *hist-piNTCF*. The variables analysed are: (a, d) potential temperature (*thetao*), (b, e) salinity (*so*) and (c, f) potential density with reference pressure of 0 dbar (*sigma0*). Colours represent data from individual models (BCC-ESM1: blue, EC-Earth3-AerChem: green, MRI-ESM2-0: yellow, UKESM1-0-LL: pink). Each model mean (thick lines) is obtained from 3 different members (thin lines). EC-Earth3-AerChem members r(3,4)i1pif1 are dashed to represent the collapsed convection.

contribution. However, the relatively small seasonal variability of the haline contribution, as well as the larger magnitude of the summer thermal signal, suggest that temperature anomalies are the dominant destabilising factor, while salinity anomalies reinforce and sustain convection.

To contextualise these findings, previous studies consistently report that aerosols significantly impact convection in the Labrador Sea (Menary et al., 2020; Liu et al., 2024). Hassan et al. (2021) propose that aerosols affect SPNA surface densities through direct changes in radiation and temperature as well as through the modification of pressure and wind patterns, leading to changes in latent and sensible heat fluxes. Moreover, Robson et al. (2022) find that aerosols enhance convection in the SPNA primarily through the advection of continental cold anomalies that increase turbulent heat loss, combined with a direct local reduction in shortwave radiation. Both studies emphasize the presence of AMOC-related feedbacks, where changes in salinity and oceanic poleward heat transport further amplify the convection response.

The detected Labrador Sea cooling in response to NTCFs likely reflects the influence of aerosol negative forcing. This interpretation is further supported by previous studies that link aerosol forcing to enhanced ocean convection in this region. However, the precise mechanisms – whether through direct local forcing or via remote advection of aerosol-induced anomalies – remain undetermined. Additionally, we cannot rule out the potential influence of sea ice changes, given that increased NTCF concentrations are associated with greater sea ice extent in the Labrador Sea (Fig. B3).

## 3.4 NTCFs impact on tropical precipitation

The impact of NTCFs on the ITCZ is evaluated using the zonal mean precipitation centroid from 20°S to 20°N (see subsection 2.4). The centroid coordinates (lat, pr) serve as indicators of the ITCZ's latitudinal position and precipitation amount across the *historical* and *hist-piNTCF* ensembles (Fig. 7). Our results indicate that historical NTCFs lead to a southward shift of the ITCZ and a net reduction in equatorial precipitation, both signals remaining consistent over the study period and across models. The latitudinal response in the individual models, represented by the *historical* and *hist-piNTCF* difference (Fig. 7b), exhibits multi-decadal oscillations, particularly prominent in MRI-ESM2-0 and BCC-ESM1. However, no evidence of a robust long-term trend emerges. In contrast, the equatorial rainfall decline due to NTCFs progressively intensifies in all models (Fig. 7d).

To quantify these responses, we define two metrics. First, the latitudinal response (latR) measures the mean ITCZ latitude difference between ensembles over 1950–2014 (Eq. (12)):

$$latR\,(°) = \overline{lat}_{historical} - \overline{lat}_{hist-piNTCF} \tag{12}$$

This calculation reveals a mean southward ITCZ displacement of 0.6° due to historical NTCFs in the ensemble mean (see Table A4 for individual model values). Second, the precipitation response (prR) captures the percentage change in ITCZ precipitation over time, using a linear approximation (Eq. (13)):

$$prR\,(\%) = \frac{mN}{pr_{clim}} \times 100 \tag{13}$$

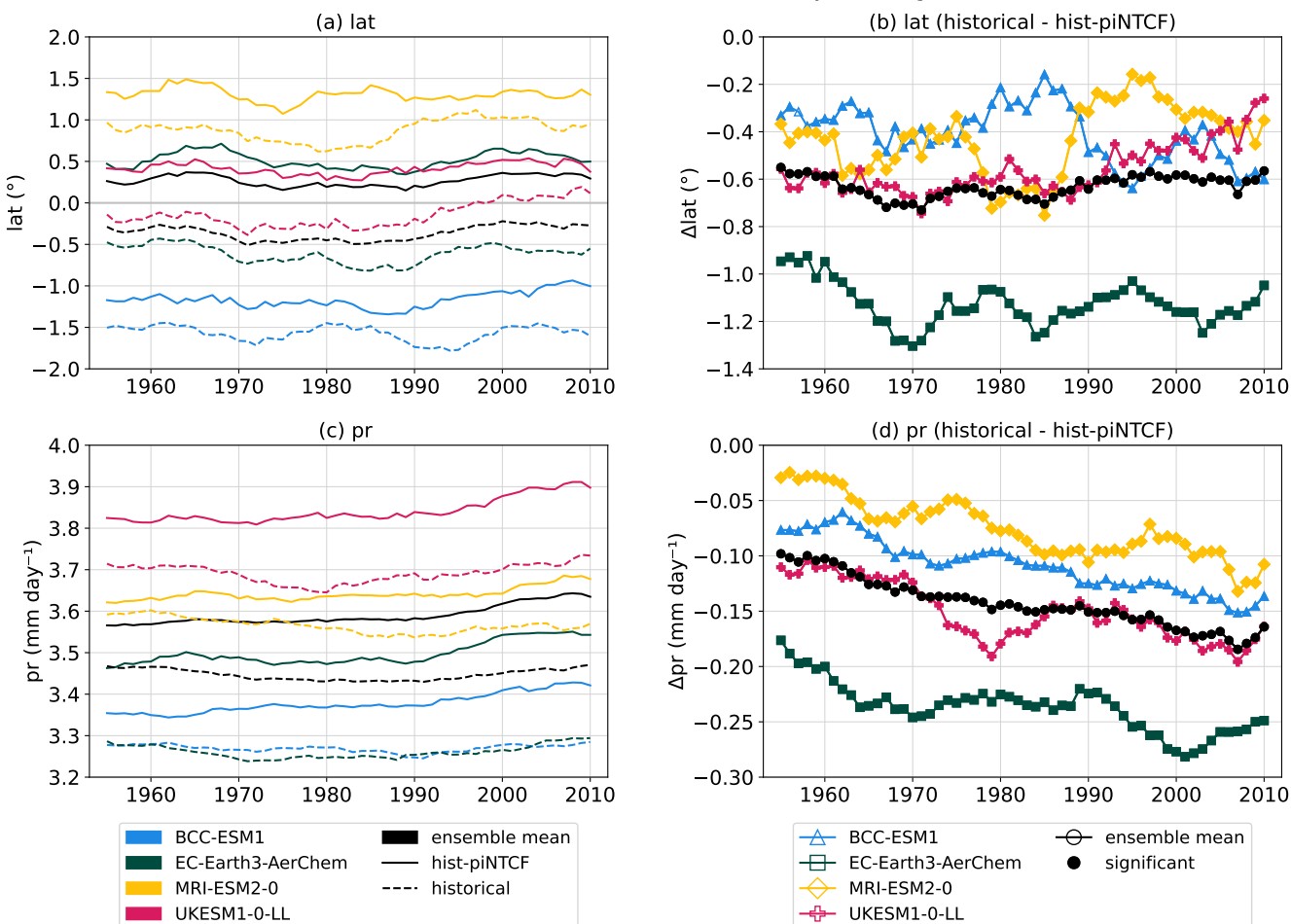

**Figure 7.** Impact of NTCFs on ITCZ latitude (a, b) and intensity (c, d), as derived from the comparison of *historical* and *hist-piNTCF* CMIP6 simulations during 1950–2014. (a, c) Ten-year rolling mean of ITCZ indices (*lat* and *pr*) for *historical* (solid) and *hist-piNTCF* (dashed) experiments and (b, d) the experiment difference, where filled symbols indicate significant values based on a bootstrapping significance test with 95% confidence (see subsection 2.2). Indices are based on the zonal average precipitation centroid from 20°S to 20°N. Colours represent data from individual models (BCC-ESM1: blue triangles, EC-Earth3-AerChem: green squares, MRI-ESM2-0: yellow diamonds, UKESM1-0-LL: pink crosses) and black circles show the model mean. For each experiment and model we consider the mean of 3 members.

where $pr_{clim}$ represents the mean ITCZ precipitation in the first decade of *hist-piNTCF*, $m$ is the slope of the precipitation difference between *historical* and *hist-piNTCF* (Fig. 7d), and N equals 65 years. Over 1950–2014, the multi-model ensemble mean shows a 2.0% weakening of the ITCZ due to historical NTCFs (see Table A4 for individual model values).

To explore the relationship between radiative fluxes and ITCZ shifts, we introduce the Net Radiation Hemispheric Difference index (*netR_HD*; Fig. B7a). This index quantifies the difference in mean net radiation at the top of the atmosphere (in terms

of CMIP6 variables: *rsdt - rsut - rlut*) between the Southern and Northern Hemispheres (SH - NH), similar to metrics used

in previous studies (Menary et al., 2020; Robson et al., 2022). Figure 8 shows that higher *netR_HD* values correlate with more southern ITCZ latitudes, suggesting that interhemispheric radiation changes produce latitudinal circulation shifts. The southward displacement of the ITCZ exhibited by the *historical* ensemble could be a response to the decrease in radiative energy in the NH, partially compensating for the interhemispheric heat imbalance.

The hemispheric radiation response closely correlates with aerosol distributions, as revealed by the Aerosol Optical Depth

Hemispheric Difference index (*od550aer_HD*; Fig. B7b). This index, analogous to *netR_HD*, uses the aerosol optical depth at 550 nm wavelength (*od550aer*) as a proxy for columnar aerosol concentration. Therefore, *od550aer_HD* represents the

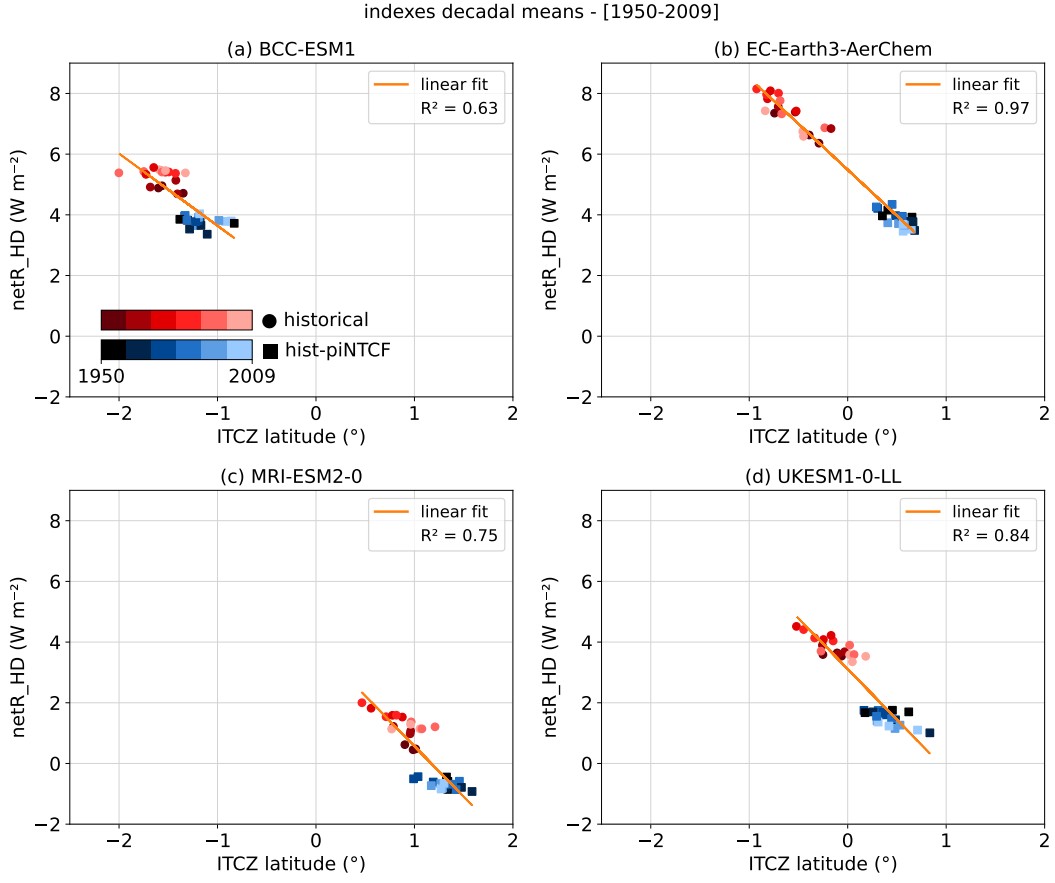

**Figure 8.** Relationship between the net radiation hemispheric difference index (*netR_HD*) and the ITCZ latitude index in two multi-model ensembles of *historical* and *hist-piNTCF* CMIP6 simulations. Decadal means of the indexes over the period 1950–2009 are represented by red dots (*historical*) and blue squares (*hist-piNTCF*), with lighter shades indicating more recent decades. Data from 3 members is plotted for each experiment and model ((a) BCC-ESM1, (b) EC-Earth3-AerChem (c) MRI-ESM2-0, (d) UKESM1-0-LL). The orange line denotes the linear fit across both ensembles.

difference in mean *od550aer* between the hemispheres (SH - NH). Figure 9 shows that higher *netR_HD* values generally correspond to lower *od550aer_HD* values, implying that a greater NH aerosol burden enhances interhemispheric radiative imbalances (i.e., less absorbed radiation in the NH and/or more absorbed radiation in the SH). These results highlight the crucial role of aerosols in modulating the interhemispheric radiation balance, which in turn can explain the ITCZ displacements.

Interestingly, EC-Earth3-AerChem and UKESM1-0-LL show a distinct behaviour in the later decades: *od550aer_HD* remains nearly unchanged after the 1980s as global aerosol concentrations stabilise, whereas *netR_HD* decreases (Fig. 9). This suggests a declining effectiveness of aerosols in offsetting GHG-induced radiative changes (Bauer et al., 2022), explaining the weakening correlation between *od550aer_HD* and *netR_HD*.

To further contextualise the aerosol–radiation connection, Figure B8 shows a spatial alignment between the net radiation and cloud forcing (difference between all-sky and clear-sky net radiation) weaker over polar regions, highlighting the relevance of clouds in shaping the radiative response to NTCFs. The global mean cloud radiative forcing (-0.59 W m$^{-2}$) exceeds the net radiation signal (-0.42 W m$^{-2}$), which is consistent with aerosol indirect negative forcing being partially compensated by the positive forcing of absorbing aerosols and tropospheric ozone. Although our analysis does not isolate individual forcing pathways this interpretation agrees with estimates of NTCFs effective radiative forcing (ERF) in previous studies. Thornhill et al. (2021) analysing AerChemMIP output found that NTCF ERF (-0.89 $\pm$ 0.20 W m$^{-2}$) is dominated by aerosols (-1.01 $\pm$ 0.25 W m$^{-2}$) over tropospheric ozone (0.03 $\pm$ 0.01 W m$^{-2}$) and its precursors (0.20 $\pm$ 0.07 W m$^{-2}$). Also using CMIP6 data,

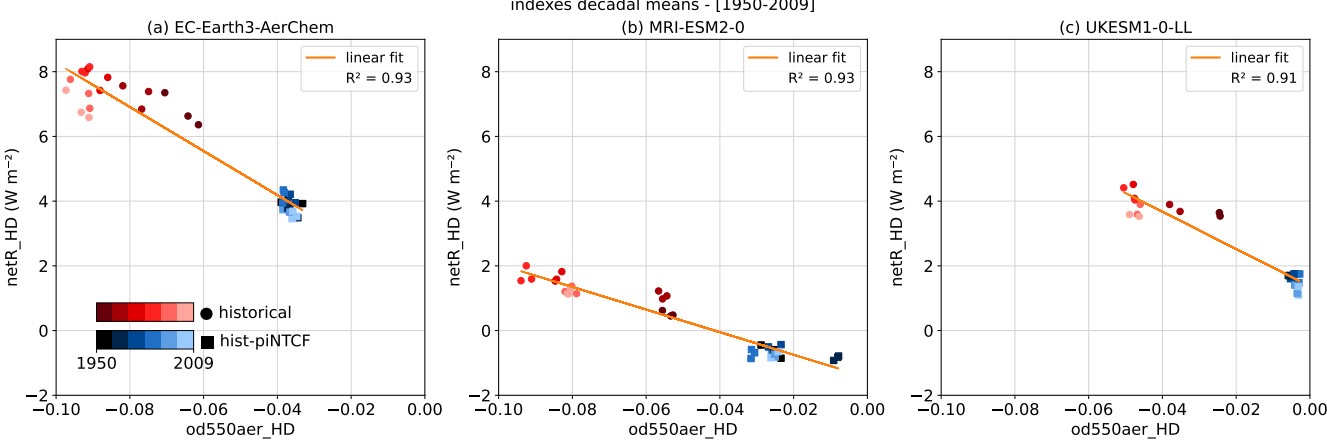

**Figure 9.** Relationship between the net radiation hemispheric difference index (*netR_HD*) and the aerosol optical depth hemispheric difference index (*od550aer_HD*) in the ensembles of *historical* and *hist-piNTCF* CMIP6 simulations. Decadal means over the period 1950–2009 are represented by red dots (*historical*) and blue squares (*hist-piNTCF*), with lighter shades indicating more recent decades. Data from 3 members is plotted for each experiment and model except for UKESM1-0-LL (c), which has *od550aer* data available from only two members, and BCC-ESM1 with no data available for this variable. Note that MRI-ESM2-0 (b) resolves stratospheric chemistry and therefore stratospheric aerosols are included in the *od550aer* variable. Regardless, all models in this study account for the radiative effects of volcanic aerosols either explicitly or through prescribed datasets or parameterisations. The orange line denotes the linear fit across both ensembles.

Forster et al. (2021) gave an estimation of aerosol ERF (-1.11 $\pm$ 0.38 W m$^{-2}$) and found a prevalence of the aerosol–cloud interaction component (-0.86 $\pm$ 0.57 W m$^{-2}$) with respect to aerosol–radiation interactions (-0.25 $\pm$ 0.40 W m$^{-2}$). Aerosol–cloud interactions therefore are a dominant contributor to NTCF forcing, but also constitute a major source of inter-model uncertainty (Szopa et al., 2021; Zhang et al., 2021). While ERF values are not directly comparable to our coupled-simulation signals, they provide useful context and share the hypothesis that clouds play a central role in the radiative forcing of NTCFs.

Changes in total cloud cover (*clt*) also broadly align with the net radiation signal, particularly over mid to high latitudes (Fig. B8c). The weaker spatial correspondence compared to that of cloud forcing (Fig. B8b) suggests that NTCFs affect not only cloud amount but also cloud properties. On the global scale, we obtain a mean increase in *clt* of +0.5% due to historical NTCFs, comparable to the values reported by Zhang et al. (2021) for aerosol forcing. Consistent with Andersen et al. (2023), we find an inverse relationship between mid-latitude clouds and absorbed radiation due to the dominance of short-wave (SW) reflection (white stippling), while in tropical regions cloud long-wave (LW) retention prevails (black stippling). Additional SW–LW decomposition of the radiative fluxes (not shown) supports this interpretation. In the Northern Hemisphere, regions with high anthropogenic NTCF emissions–such as North America, Europe and East Asia–and their downwind regions exhibit both increased cloud cover and, more strongly, negative cloud radiative forcing dominated by the SW component. This, combined with the more localised clear-sky SW reflectivity increases, produces a predominantly zonal negative signal. In the Southern Hemisphere, clear-sky fluxes reveal a dominant role of the LW component resulting in a positive hemispheric signal, reinforced by cloud forcing: in the tropics, the increase in cloud cover leads to higher LW retention , whereas the decrease in mid-latitude cloud cover likely reduces SW reflectivity. Altogether, these results underline that cloud radiative forcing is a key contributor to the interhemispheric radiative imbalance previously identified. Further insight into the nature of the hemispheric asymmetry response to NTCFs is its agreement with the spatial pattern of aerosol forcing since 1850 reported by Szopa et al. (2021), which shows the strongest negative ERF over and downwind of industrialized regions in the Northern Hemisphere.

Analogously to the ITCZ latitude analysis, we studied the relationship between equatorial rainfall amount (pr centroid coordinate) and the *netR_HD* index (Fig. B9). The analysis reveals a clear negative correlation, with higher *netR_HD* values associated with reduced tropical rainfall. Because NTCFs exert negative radiative forcing (Fig. B8a) and subsequently induce surface cooling (Fig. 1b) predominantly in the Northern Hemisphere, *netR_HD* is inherently correlated with the global temperature decrease. Therefore, Figure B9 could be reflecting the expected reduction in precipitation that accompanies the global cooling (Held and Soden, 2006). Additional analysis confirms this temperature–precipitation link, but we find it to be weaker than the relationship between precipitation and *netR_HD* (not shown). While the physical interpretation of this relation is complex, our results suggest that the tropical precipitation response to NTCFs likely combines the effects of global cooling and atmospheric circulation changes driven by the interhemispheric radiation imbalance. As shown by Allen and Sherwood (2011), such an imbalance can weaken the Hadley cell, which could contribute to the precipitation reduction. Additionally, the southward displacement of the ITCZ (Fig. 7b) may further modulate rainfall by shifting convection zones between oceanic and continental regions.

Overall, our results suggest that the historical climate response to anthropogenic NTCFs was largely shaped by aerosol radiative forcing. While we did not explicitly isolate the effects of individual species, the net climate signal appears to be

primarily driven by negative radiative forcing, despite NTCFs with opposing effects (e.g., absorbing aerosols and tropospheric ozone). Additionally, the emerging signals align with previous studies that report comparable temperature and circulation changes in response to aerosols. Beyond describing these effects, we provide quantitative estimates of regional climate impacts of NTCFs, spanning the Arctic, North Atlantic and tropical systems, across multiple CMIP6 ESMs with different chemistry schemes.

## 4    Conclusions

NTCFs play a crucial role in shaping the climate system by offsetting the warming effects of GHG, amongst other impacts. While air quality mitigation necessarily aims to reduce NTCF concentrations due to their direct impacts on health and ecosystems, it is essential to understand their broader climatic implications. By focusing on the recent historical period, this study assess the role of NTCFs within their evolving concentrations with time, providing insights into their importance relative to other anthropogenic forcings.

In this study, we investigated the impact of NTCFs on the global climate system using a multi-model ensemble of CMIP6 simulations. By comparing the *historical* and *hist-piNTCF* experiments, we isolated the effects of NTCFs from those of other anthropogenic and natural forcings. Our analysis reveals three primary climatic signals induced by NTCF changes:

- **Amplified Arctic Cooling:** The presence of NTCFs leads to global cooling, suggesting the dominance of species with negative radiative forcing such as aerosols. This effect is particularly strong in the Arctic region, where the seasonality of the cooling signal and the associated sea ice response across models indicate an active role of Arctic Amplification feedback mechanisms. The multi-model means show that during 1950–1980, historical NTCFs cooled the Arctic nearly four times more than the rest of the globe. After the 1980s, the relevance of aerosol forcing diminished, with GHGs dominating both the global and Arctic temperature trends.

- **Increased Labrador Sea Convection:** Over the period 1950–2014, historical NTCFs contributed to a 38% increase in FMA Labrador Sea convection. This increase is consistent with summer surface cooling eroding water-column stratification and favouring deeper convection in the following months. Once convection is active, salinity anomalies appear to lead the density response.

- **Southward Displacement of the ITCZ:** Higher concentrations of NTCFs in the Northern Hemisphere, specifically aerosols, result in an interhemispheric radiation imbalance. This latitudinal gradient leads to an ITCZ southward displacement of 0.6°, with decreased precipitation north of the equator and increased precipitation to the south of it. Changes in the tropical rain belt contribute to an overall 2.0% weakening of tropical precipitation (20°N–20°S) in the multi-model mean.

These findings highlight the historical relevance of NTCFs and point to key research directions:

- **Arctic Climate:** While our results suggest that NTCFs–particularly aerosols–have mitigated Arctic warming to some extent, the relative roles of different mechanisms—such as aerosol–radiation interactions, aerosol–cloud interactions,

and indirect changes in energy transport—remain to be quantified. Disentangling these contributions will be crucial for improving projections of future Arctic climate change. Additionally, further studying the influence of NTCFs on Arctic Amplification feedback mechanisms, which strongly depend on sea ice evolution, is critical given the projected decline in Arctic sea ice.

- **North Atlantic Circulation:** The impact on Labrador Sea convection may have broader implications for the subpolar gyre and the AMOC. Further work is envisaged on the origin of the identified surface cooling in the Labrador Sea and the mechanisms at play, using a large ensemble of scenario simulations from the AerChemMIP initiative.

- **Tropical Precipitation:** Despite known uncertainties in ITCZ representation across models (Tian and Dong, 2020), our analysis revealed a consistent circulation shift in response to NTCFs. Understanding whether similar responses would emerge under future reductions in NTCFs is crucial, as ITCZ variability strongly influences equatorial precipitation patterns, with potential consequences for monsoon systems and regional hydrological cycles.

Our analysis relies on a limited subset of Earth System Models, which may introduce model-specific biases – including those associated with coarse spatial resolution and parameterised ocean eddies (Hallberg, 2013). Despite this inter-model uncertainty, we have successfully quantified the climate impact of NTCFs with statistically significant climate signals across models. Nonetheless, to refine our understanding of the effects of individual NTCF species (e.g., different aerosol types, ozone, and their precursors) as well as their regional influences, we advocate for expanding the CMIP ensemble of attribution-focused simulations.

This study highlights the substantial role of NTCFs in shaping past climate on both global and regional scales. We delve into the mechanisms driving climatic changes and emphasise the importance of incorporating interactive NTCFs in climate projections. While anthropogenic NTCFs will likely decline in the future, natural aerosols' evolution remains uncertain. Hence it is essential to understand the feedbacks between natural and anthropogenic species, as well as their evolving contributions under changing climatic conditions, and to accurately quantify their effects to shape effective GHG mitigation strategies.

*Code and data availability.* The analyses developed in this study use CMIP6 data, publicly available on the Earth System Grid Federation (ESGF) portal. Specific code will be uploaded on the final version of the article. If needed for the reviewing process, it will be provided upon request.

**Appendix A:  Supplementary tables**

**Table A1.** Ensemble members and components of the models used in the study.

| model | BCC-ESM1 | EC-Earth3-AerChem | MRI-ESM2-0 | UKESM1-0-LL |
|---|---|---|---|---|
| **members** | r(1:3)i1p1f1 | r(1,3,4)i1p1f1 | r(1,3,5)i1p1f1 | r(1:3)i1p1f1 |
| **atmosphere** | BCC-AGCM3-Chem | IFS | MRI-AGCM3.5 | MetUM-HadGEM3-GA7.1 |
| resolution (lat × lon × lev) | 2.8° × 2.8° × 26L | 0.7° × 0.7° × 91L | 1.125° × 1.125° × 80L | 1.25° × 1.875° × 85L |
| top level | 2.914 hPa | 0.01 hPa | 0.01 hPa | 85 km |
| **aerosols & reactive gases** | Same as atmosphere | TM5-mp 3.0 | aerosols: MASINGAR mk-2r4c<br>gases: MRI-CCM2.1 | UKCA-StratTrop |
| resolution (lat × lon × lev) | | 2° × 3° × 34L | 0.938° × 1.875° × 80L<br>1.4° × 2.8° × 80L | 1.25° × 1.875° × 85L |
| **ocean** | MOM4-L40 | NEMO 3.6 | MRI.COM4.4 (+bgchem) | NEMO-HadGEM3-GO6.0 + MEDUSA2(bgchem) |
| resolution (lat × lon × lev) | 1/3° × 1° × 40L (30S-30N)<br>1° × 1° × 40L (rest) | 1/3° × 1° × 75L (30S-30N)<br>1° × 1° × 75L (rest) | 0.3° × 1° × 61L (10S-10N)<br>0.5° × 1° × 61L (rest) | 1/3° × 1° × 75L (30S-30N)<br>1° × 1° × 75L (rest) |
| **sea ice** | SIS2 | LIM3 | Same as ocean | CICE-HadGEM3-GSI8 |
| **land** | BCC-AVIM2.0 | HTESSEL | HAL 1.0 | JULES-ES-1.0 |

**Table A2.** Arctic Amplification Factor (AAF; Wu et al., 2024) attributed to Near-Term Climate Forcers (NTCFs) as derived from the comparison of *historical* and *hist-piNTCF* CMIP6 simulations over 1950–1980. Values shown for individual models and their ensemble mean.

| Model | AAF pre80s |
|---|---|
| BCC-ESM1 | 3.85 |
| EC-Earth3-AerChem | 4.48 |
| MRI-ESM2-0 | 3.12 |
| UKESM1-0-LL | 4.10 |
| Ensemble Mean | 3.87 |

**Table A3.** Labrador Sea Convection Response (LSCR; percentage change in mixed layer depth) attributed NTCFs as derived from the comparison of *historical* and *hist-piNTCF* CMIP6 simulations over 1950–2014. Values shown for individual models and their ensemble mean.

| Model | LSCR (%) |
|---|---|
| BCC-ESM1 | 24.01 |
| EC-Earth3-AerChem | 59.66 |
| MRI-ESM2-0 | 25.98 |
| UKESM1-0-LL | 194.23 |
| Ensemble Mean | 37.82 |

**Table A4.** ITCZ response to NTCFs quantified by latitude displacement ($\Delta$lat) and precipitation change ($\Delta$pr) as derived from the comparison of *historical* and *hist-piNTCF* CMIP6 simulations over 1950–2014. Values shown for individual models and their ensemble mean.

| Model | $\Delta$lat(°) | $\Delta$pr(%) |
|---|---|---|
| BCC-ESM1 | -0.4 | -2.5 |
| EC-Earth3-AerChem | -1.1 | -1.8 |
| MRI-ESM2-0 | -0.4 | -1.8 |
| UKESM1-0-LL | -0.6 | -1.9 |
| Ensemble Mean | -0.6 | -2.0 |

## Appendix B: Supplementary figures

### B1   NTCFs impact on Arctic temperature

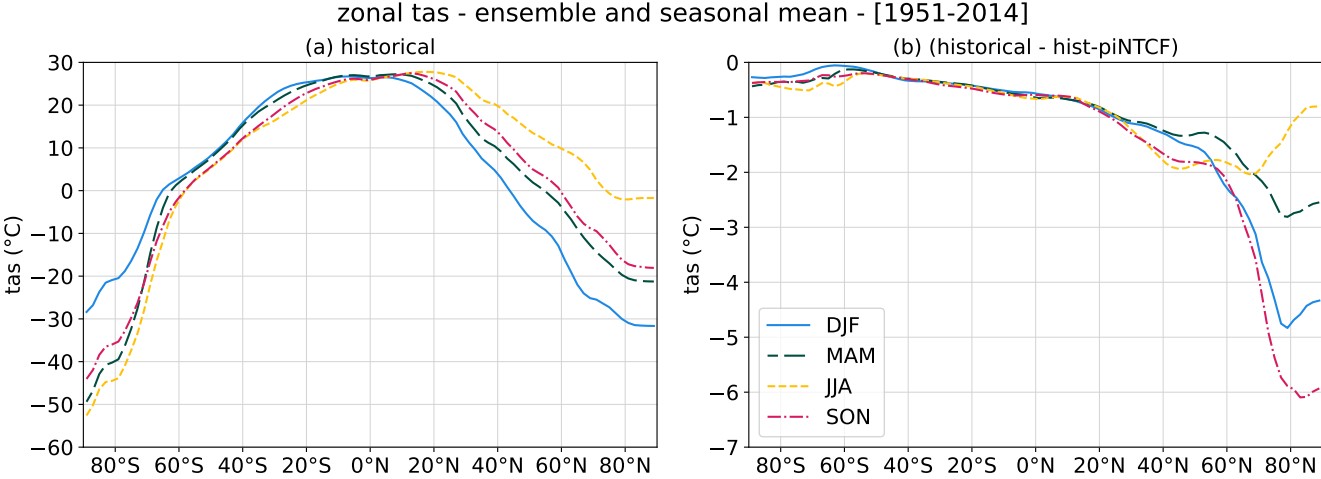

**Figure B1.** Impact of historical NTCF forcings on seasonal zonal surface air temperature (*tas*) during the period 1951-2014. (a) Seasonal climatology (DJF: blue, MAM: green, JJA: yellow, SON: pink) for the multi-model *historical* mean and (b) difference in climatologies between the multi-model *historical* and *hist-piNTCF* ensemble means. The ensembles analysed are comprised of 4 models (BCC-ESM1, MRI-ESM2-0, UKESM1-0-LL and EC-Earth3-AerChem) with 3 members each.

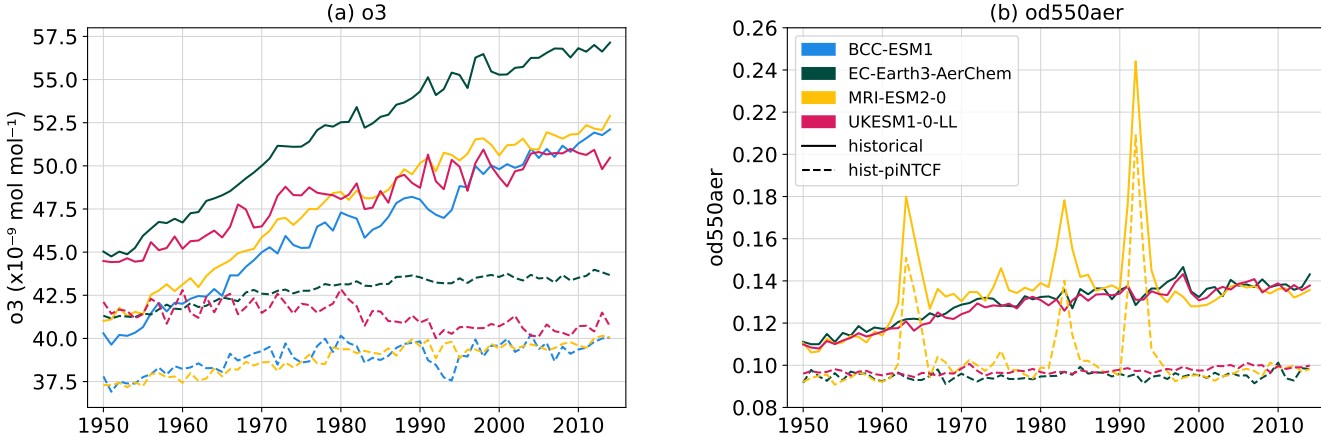

**Figure B2.** Annual global mean evolution of the (a) ozone concentration (*o3*) at 500 hPa and (b) aerosol optical depth at 550nm (*od550aer*) for the *historical* (solid lines) and *hist-piNTCF* (dashed lines) CMIP6 simulations over the period 1950–2014. Colours represent data from individual models (BCC-ESM1: blue, EC-Earth3-AerChem: green, MRI-ESM2-0: yellow, UKESM1-0-LL: pink). Each model mean is obtained from 3 different members but for UKESM1-0-LL *od550aer* data, which only has 2 members available, and BCC-ESM1 with no data available for this variable. Note that MRI-ESM2-0 resolves stratospheric chemistry and its effect is included in the *od550aer*. As a result, peaks following major volcanic eruptions are present. Regardless, all models in this study account for the radiative effects of volcanic aerosols either explicitly or through prescribed datasets or parametrisations.

## siconc SON [1950-2014]

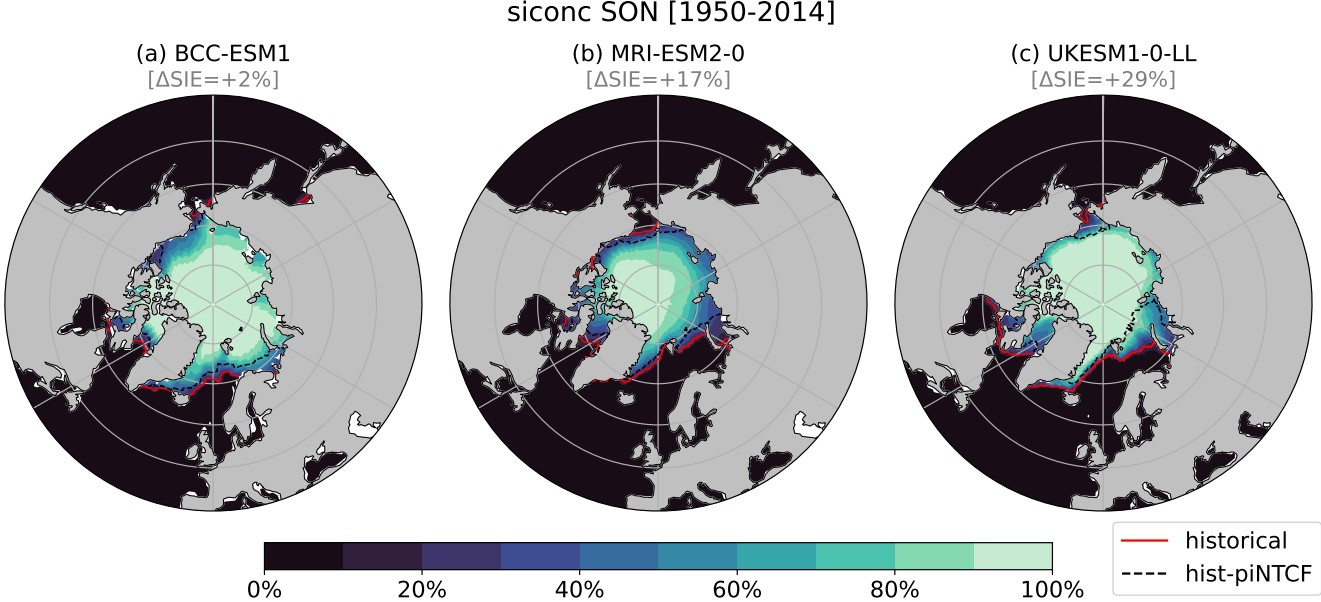

**Figure B3.** Impact of historical NTCFs on mean sea ice concentration (*siconc*) in boreal autumn (September, October and November) during the period 1951–2014 for different CMIP6 models ((a) BCC-ESM1, (b) MRI-ESM2-0, (c) UKESM1-0-LL). The colours represent the *siconc* climatology for the *historical* experiment and the contours the sea ice edge (*siconc* = 15%) for the experiments *historical* (solid red) and *hist-piNTCF* (dashed black). Values of sea ice extent change (ΔSIE, shown in gray above each panel) denote the percentage difference in total area with *siconc* ≥ 15% (sea ice extent) between *historical* and *hist-piNTCF* experiments, relative to the *hist-piNTCF* extent. For each experiment and model we consider the mean of 3 members but for BCC-ESM1 *hist-piNTCF*, with data available only for 1 member. Note that EC-Earth3-AerChem is not shown as *siconc* data were not available for the *hist-piNTCF* experiment.

## B2   NTCFs impact on Labrador Sea convection

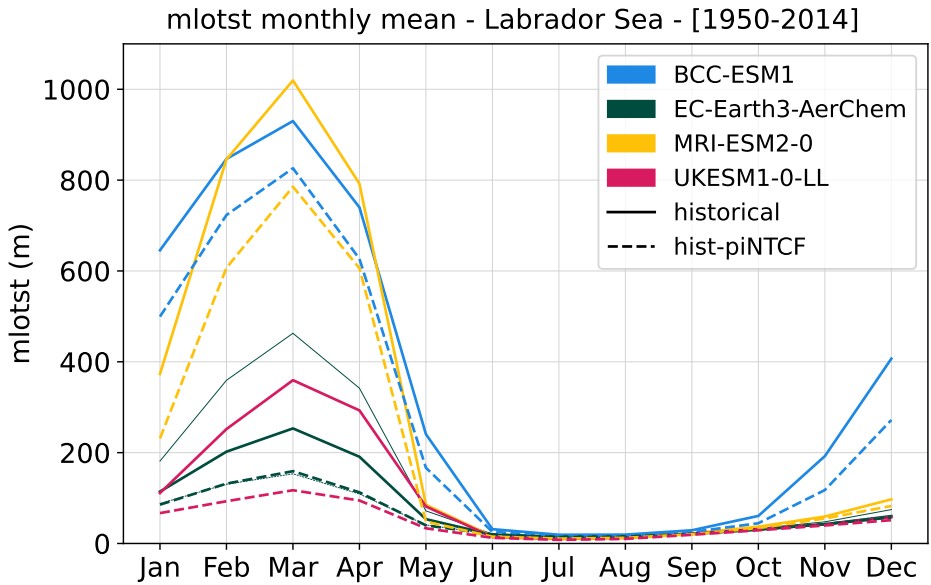

**Figure B4.** Seasonal cycle of the Labrador Sea ocean mixed layer thickness defined by sigma T (*mlotst*), as derived from the *historical* (solid lines) and *hist-piNTCF* (dashed lines) CMIP6 simulations over the period 1950–2014. Colours represent data from individual models (BCC-ESM1: blue, EC-Earth3-AerChem: green, MRI-ESM2-0: yellow, UKESM1-0-LL: pink). Each model mean (thick lines) is obtained from 3 different members. The EC-Earth3-Aerchem model member r1i1pif1 is represented as well (thin lines). The Labrador Sea area is defined as: (60°W, 45°W; 50°N, 65°N).

**Figure B5.** Comparison of the Labrador Sea ocean mixed layer thickness defined by sigma T (*mlotst*) for the different members of the model EC-Earth3-Aerchem. Data is obtained from the CMIP6 simulations: (a) *historical* and (b) *hist-piNTCF*. The period of study considers the months of February, March and April (FMA) between 1950–2014. The Labrador Sea area is defined as: (60°W, 45°W; 50°N, 65°N).

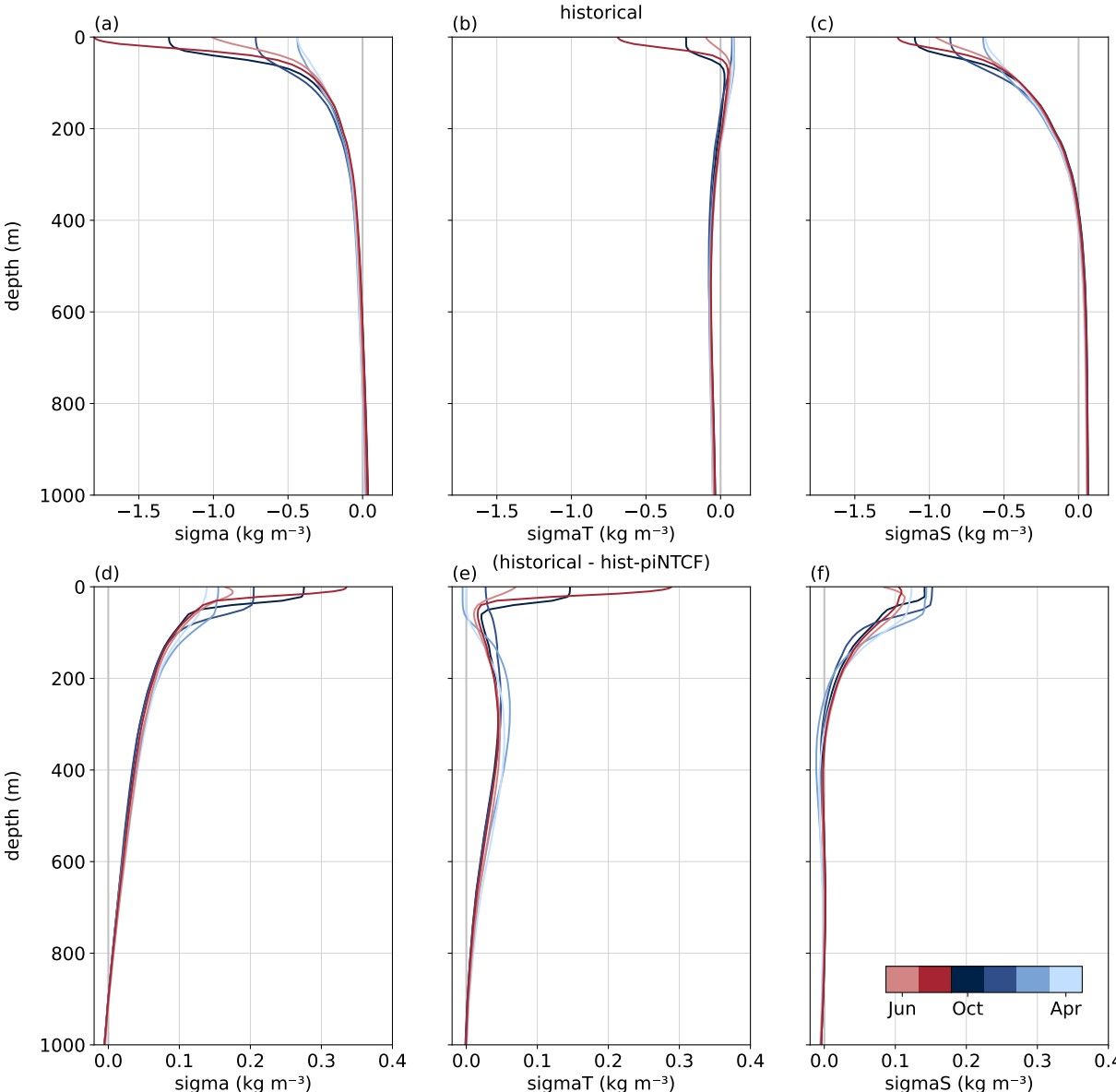

**Figure B6.** Impact of NTCFs on the contributions of temperature and salinity to density changes in the Labrador Sea (60°W, 45°W; 50°N, 65°N), based on the comparison of *historical* and *hist-piNTCF* CMIP6 simulations over the period 1950–2014. (a, b, c) Monthly climatology for the *historical* experiment, and (d, e, f) mean difference between *historical* and *hist-piNTCF*. The variables analysed are: (a, d) potential density anomalies (*sigma*) (b, e) temperature's contribution to density (*sigmaT*) and (c, f) salinity's contribution to density (*sigmaS*). Details on the computation of these magnitudes are provided in subsection 2.3. To enhance clarity only data from every second month is shown. Colours represent the state of climatological convection according to Figure B4: active (Oct-May; blue) and nonactive (Jun-Sep; red). The ensembles analysed are comprised of 4 models (BCC-ESM1, MRI-ESM2-0, UKESM1-0-LL and EC-Earth3-AerChem), with 3 members each but for EC-Earth3-AerChem, which only has one member without suppressed convection.

## B3    NTCFs impact on tropical precipitation

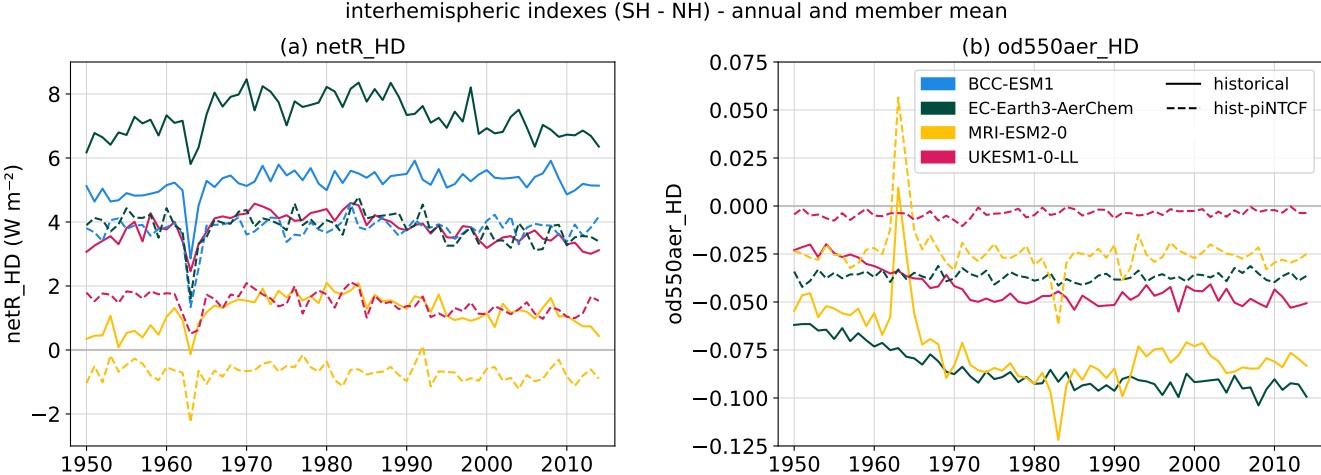

**Figure B7.** Annual mean evolution of the (a) *netR_HD* and (b) *od550aer_HD* indexes for the *historical* (solid lines) and *hist-piNTCF* (dashed lines) CMIP6 simulations over the period 1950–2014. Colours represent data from individual models (BCC-ESM1: blue, EC-Earth3-AerChem: green, MRI-ESM2-0: yellow, UKESM1-0-LL: pink). Each model mean is obtained from 3 different members but for UKESM1-0-LL *od550aer* data, which only has 2 members available, and BCC-ESM1 with no data available for this variable. Note that MRI-ESM2-0 resolves stratospheric chemistry and its effect is included in the *od550aer*. As a result, peaks following major volcanic eruptions are present. Regardless, all models in this study account for the radiative effects of volcanic aerosols either explicitly or through prescribed datasets or parametrisations.

Ensemble mean (historical - hist-piNTCF) - [1950-2014]

a) ΔnetR    [-0.42 W m⁻²]
b) Δcloud forcing    [-0.59 W m⁻²]
c) Δclt    [0.5 %]

α = 0.05    significant    both significant and correlated    both significant and anticorrelated

**Figure B8.** Impact of historical NTCF forcings on (a) net radiation at the top of the atmosphere (*netR*), (b) cloud forcing (all-sky minus clear-sky *netR*) and (c) cloud cover (*clt*), as derived from the comparison of *historical* and *hist-piNTCF* CMIP6 simulations over the period 1950–2014. The panels show the difference in climatologies between the multi-model *historical* and *hist-piNTCF* ensemble means. Global mean values for each magnitude are shown in gray in the titles. Both ensembles are comprised of four models (BCC-ESM1, MRI-ESM2-0, UKESM1-0-LL, and EC-Earth3-AerChem), with three members each. Stippling colours indicate whether the signal of the represented variable is significant (grey), both variables are significant (black) or both variables are significant and have opposing sign (white). Significance is determined through a paired sample t-test with 95% confidence.

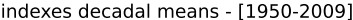

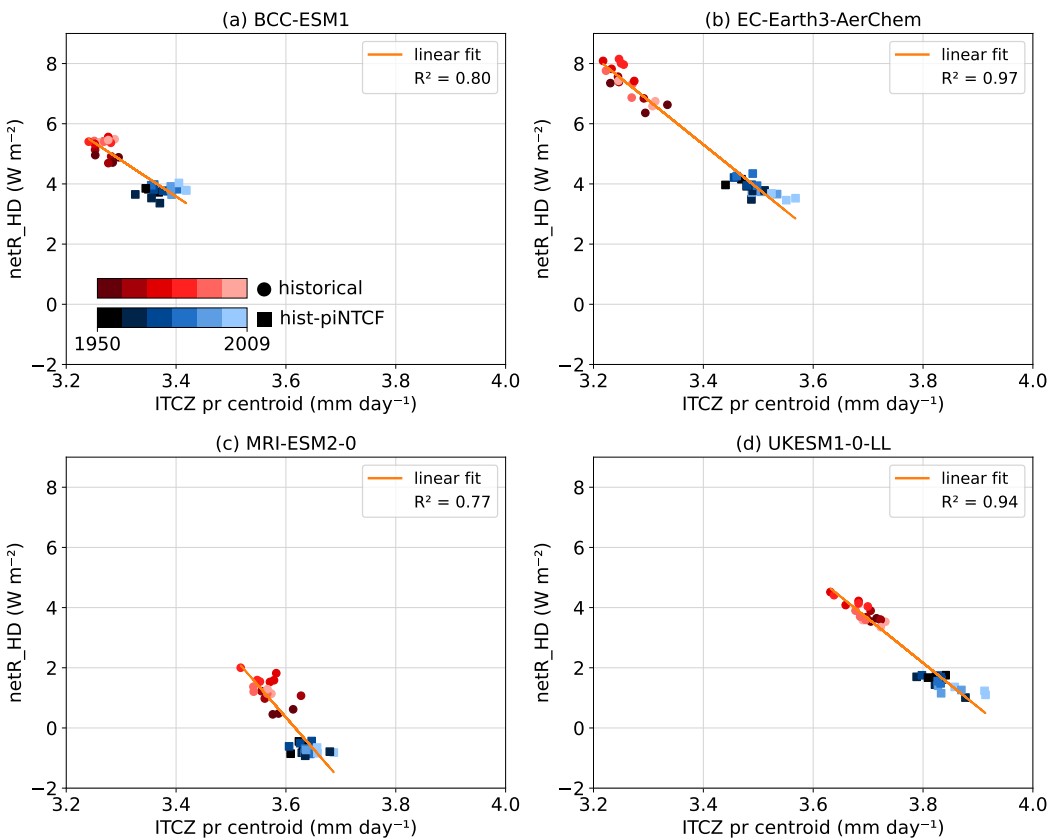

**Figure B9.** Relationship between the net radiation hemispheric difference index (*netR_HD*) and the ITCZ precipitation amount index in two multi-model ensembles of *historical* and *hist-piNTCF* CMIP6 simulations. Decadal means of the indexes over the period 1950–2009 are represented by red dots (*historical*) and blue squares (*hist-piNTCF*), with lighter shades indicating more recent decades. Data from 3 members is plotted for each experiment and model ((a) BCC-ESM1, (b) EC-Earth3-AerChem (c) MRI-ESM2-0, (d) UKESM1-0-LL). The orange line denotes the linear fit across both ensembles.

*Author contributions.* ASE formal analysis, visualisation and writing. MGA and PO conceptualisation, supervision and writing review. MSC data curation. SLT software. CPG and MGD writing review.

*Competing interests.* The authors declare that they have no conflict of interest.

*Acknowledgements.* We extend our gratitude to Eneko Martin-Martinez, Aude Carréric and Roberto Bilbao, for their constructive feedback on the analyses performed. We thank the in-house technical support group easing our everyday work. We also appreciate the ESMValTool development team for their work and tool support.

We acknowledge the World Climate Research Programme, which, through its Working Group on Coupled Modelling, coordinated and promoted CMIP6. We thank the climate modeling groups for producing and making available their model output, the Earth System Grid Federation (ESGF) for archiving the data and providing access, and the multiple funding agencies who support CMIP6 and ESGF.

The research leading to these results has received funding from the EU HE Framework Programme under grant agreement N° GA 101056783 and 101137680 (FOCI and CERTAINTY) and the AXA Research Fund through the AXA Chair on Sand and Dust Storms at BSC. Furthermore, we thank the support of the Generalitat de Catalunya Department of Research and Universities to the Research Groups CVC and AC (Codes: 2021 SGR 00786 and 01550)

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
