# Peer review of "Regional climate imprints of recent historical changes in anthropogenic Near Term Climate Forcers"

_EGUsphere, 2025_

## Author Response (AR1)

**We greatly thank both reviewers for their feedback as we consider their insights have improved the scientific completeness of our work. In the present document you can find the detailed responses (in blue) to their comments (in black).**

**Response to Referee #1**

This paper examines the climate effects of Near Term Climate Forcers (NTCFs) using two climate model experiments from CMIP6-AerChemMIP, "historical" and "hist-piNTCF", which include time-varying and fixed pre-industrial NTCF forcings, respectively. Focus is given to three main climate responses to NTCFs: 1) Arctic-amplified global cooling, 2) increased Labrador Sea convection, and 3) changes in tropical precipitation, including a southward displacement of the ITCZ.

Overall, the paper is clear and well written, and the methodology is sound. However, there are a few occasions throughout where statements are made that are unclear and/or are unsupported by the authors' results. I discuss these in my specific comments below. Once these comments are addressed, I believe that the paper should be acceptable for publication.

We thank Referee #1 for their thorough and constructive review of our manuscript. We appreciate their positive assessment of the clarity, writing quality, and methodological soundness of our work. We have carefully considered all comments and have made revisions to address the unclear statements and better support our conclusions with the presented results. Each specific comment is addressed individually below, detailing the changes made to the manuscript.

Specific comments:

1) Lines 52-54: Since this paragraph is focused on the ocean, presumably you are talking about ocean meridional circulation and ocean heat transport here?

Indeed, Cowan and Cai (2013) assesses the response of large-scale ocean circulation to aerosols. In these lines we refer to the response of the meridional overturning ocean circulation and its consequent changes in heat transport clarifies the context of their results. We have rewritten them to accurately reflect the Cowan and Cai (2013) findings:
*"Cowan and Cai (2013), using a coupled atmosphere-ocean model, reported that non-Asian aerosols dominated the ocean response to global aerosol forcing during the 20th century, delaying the GHG-induced weakening of the meridional overturning circulation and, consistently, increasing the northward heat transport across the equatorial Atlantic."*

2) Line 87: Should this be "key metrics", not "key magnitudes"?

While we agree that the ITCZ latitude and precipitation indexes are better described as "metrics", ocean density and the temperature and salinity contributions to it are more appropriately described as physical magnitudes or diagnostics. To encompass both types of indicators accurately, we have revised the terminology:
"*In the following subsections we describe the selection of model data, the statistical metrics applied, and key diagnostics used to assess NTCF impacts on specific aspects of climate such as ocean density and the ITCZ.*"

3) Table A1 header row: First model should be BCC-ESM1, not BSC-ESM1.

Corrected.

4) Lines 191-193 and Fig. 1c,d,g,h: It might be interesting to quantify how much of the variance change between historical and hist-piNTCF is due to different multidecadal trends in these two experiments versus different interannual variability. The impact of different trends on the variance change could be quantified by comparing Fig. 1d,h (which presumably include the effects of trend differences) with the analogous figures computed using detrended time series. Generally speaking, the effects of anthropogenic aerosols (which tend to dominate the NTCF response) counteract the effects of greenhouse gases, contributing to smaller trends in *historical* compared to *hist-piNTCF*. This is consistent with the overall decrease in variance shown in Fig. 1d,h.

We thank the reviewer for this suggestion. Following their recommendation, we performed an additional analysis using linearly detrended time series to compare with the original standard deviation results (Sup.Fig. 1a-d). This detrending is meaningful for the *hist-piNTCF* experiment, where aerosol forcing is fixed to pre-industrial and temperature continuously increases throughout the study period. However, it is less effective for the *historical* experiment, where aerosol forcing is relatively stable until the 1980s and then increases, producing a distinctly non-linear effect on surface temperatures (see Fig 3a,c of the manuscript). A simple linear detrend may therefore misrepresent the variability signal.

To better separate contributions from different timescales, we also applied a 10-year Butterworth low-pass (Sup.Fig 1e,f) and high-pass filter (Sup.Fig. 1g,h). The low-pass results capture the long-term NTCF forcing and multidecadal variability, while the high-pass results highlight interannual-to-decadal changes. We find that the greatest variance changes are associated with long-term and multidecadal variability, supporting our conclusion that the atmospheric signal is connected to ocean circulation and convection changes (multidecadal timescales; Grossmann and Klotzbach, 2009). Although weaker, we also find variability increases in the high-pass analysis, particularly in the Barents Sea and inner Labrador Sea. This may be linked to sea ice extent changes (see Fig B2 of the manuscript), where reduced sea ice cover exposes the ocean to stronger interannual fluctuations.

These insights have been included in the revised version of the paper. Lines 191-193 now read:
"*Secondly, we detect an increase in* tas *variability over the Labrador and Norwegian Seas, key regions of deep water formation (Fig. 1d). This variance increase concentrates on multidecadal scales (not shown), consistent with the characteristic timescales of North Atlantic ocean circulation and convection. The detected signal aligns with changes in ocean convection due to NTCFs (Delworth and Dixon, 2006; Iwi et al., 2012), explored in Subsection 3.3.*"

[Figure]

**Annual tas standard deviation - ensemble mean**

**Supporting Figure 1.** Standard deviation in time for the multi-model historical ensemble mean (a, c, e, f) and temporal variance ratio between the *historical* ensemble mean and its *hist-piNTCF* counterpart (expressed as percentage change; b, d, f, h). (a, b) Original annual surface air temperature (tas) data, (c, d) linearly detrended data, (e, f) 10-year low-pass filtered data (g, h) and 10-year high-pass filtered data. The filtering is applied with a Butterworth approach. The *historical* and *hist-piNTCF* ensembles analysed are comprised of 4 models (BCC-ESM1, MRI-ESM2-0, UKESM1-0-LL and EC-Earth3-AerChem) with 3 members each. Stippling is applied to different percentages of ensemble members coinciding in the sign of the response (b, d, f, h).

5) Fig. B2: Should probably say something in the figure caption about why you don't show the siconc from the EC-Earth model.

We agree that this omission should be explained in the figure caption, which now includes: "*Note that EC-Earth3-AerChem is not shown as* siconc *data were not available for the* hist-piNTCF *experiment.*"

6) Lines 224-225: I would change "sea ice-albedo feedback" to "sea ice-related feedbacks" here. The albedo feedback over the Arctic mainly operates in summer, but you're showing

autumn siconc here. In the autumn, it is mainly the sea ice-insulation feedback that is acting to amplify temperature changes.

We agree with the reviewer that a more general term is appropriate given the complexity of Arctic climate processes, and have changed it as suggested. Certainly, albedo is most relevant during maximum insolation periods, controlling ocean energy absorption up until early autumn. This has direct implications to sea ice formation and ocean energy release during the analysed autumn period while also being deeply interconnected with other surface processes such as sea ice-insulation (as the reviewer notes), surface heat flux, and lapse rate feedbacks.

7) Line 230: "our results suggest" appears twice.

Corrected.

8) Lines 232-233: It's unclear what is meant here by "regional radiative changes".

We agree that this phrase was too vague and potentially confusing in the context of our analysis. Lines 232-233 now read:
"*The observed increase in sea ice extent spatially aligns with the temperature response, which could suggest the operation of sea ice–related feedback mechanisms. Other processes that have been previously invoked to explain Arctic Amplification, such as changes in atmospheric and oceanic poleward energy transports (Iwi et al., 2012; Robson et al., 2022; Needham and Randall, 2023) cloud and water vapor feedback (Goosse et al., 2018), and lapse-rate feedback (Pithan and Mauritsen, 2014) may have contributed to the pronounced regional temperature changes. Quantifying their relative influence, however, lies beyond the scope of this analysis.*"

9) Line 236: Should be "formation".

Corrected.

10) Fig. 4: Figure title indicates that the period of focus is 1980-2014, while the caption indicates 1950-2014.

Corrected.

11) Lines 245-246: First of all, should say Fig. 4d, not 4f. Corrected.
Secondly, is it certain that these episodes of collapsed convection are purely stochastic? Could there be a state (and thus forcing) dependence to them? If so, the results in Fig. 4d might not change much if you had more ensemble members. All this is to say that it might be good to soften the language a bit here, e.g., say that the response to NTCFs "may be" underestimated, rather than "is likely" underestimated.

Regarding the stochastic nature of convection collapse episodes: Meccia et al. (2023) demonstrate that EC-Earth3 models exhibit multi-centennial AMOC oscillations triggered by the accumulation of salinity anomalies in the Arctic that, when released into the North Atlantic, affect water column stability and therefore convection. They state sea ice plays a driving role in the development of the salinity anomalies and find that in future scenarios with warmer conditions there is not enough sea ice to trigger the collapsing mechanism. Thus, there is indeed a state-dependency related to the background forcing. However, evidence

from larger ensembles indicates under historical forcings the background-state does not imply the occurrence of collapsed convection. From the 15 historical members produced at the BSC with the General Circulation Model (GCM) version of EC-Earth3, only 5 members show a spontaneous collapse of the Labrador Sea mixing up to 2005, which leads to a consistently lower AMOC state (as shown in Figs. 6a and 7a of Bilbao et al., 2021). Notably, the periods of collapse differ across these members, underscoring their stochastic nature even under identical external forcing.

Nevertheless, in the simulations used in our study, collapsed convection occurs only in *historical* members, never in *hist-piNTCF* ones. This suggests that the relatively warmer climate due to the reduced presence of NTCFs (and associated sea ice changes) may prevent the collapse mechanism from operating. In this sense, NTCFs forcing does play a role in the occurrence of collapsed convection.

With respect to the chosen wording, we should clarify that our statement about underestimation refers to an implication of the used methodology rather than the nature of convection collapse itself. When a *historical* simulation exhibits collapsed convection (as seen in Figure B4a), the presence or absence of NTCFs has minimal impact on the already collapsed convection state. Comparing such a *historical* member to a *hist-piNTCF* member with active convection would erroneously suggest that NTCFs decrease convection, when in fact the difference would reflect the collapsed convection state (strongly dependent on internal variability) rather than exclusively the NTCF forcing effect. In essence, the methodology used to isolate the NTCF forcing presents itself lacking in this case.

Despite having one collapsed historical member for the whole study period and another for approximately half the period, the EC-Earth3-AerChem ensemble means still show that NTCFs enhance Labrador Sea convection (Figure 4d). Thus supporting our "likely underestimated" phrasing and our decision to discard the collapsed historical members in the following analyses.

All in all, we thank the reviewer's input and deem important a clarification. Lines 243-245 now read:
"*This behaviour is consistent with known Labrador Sea convection shutdowns in EC-Earth3-models that can persist for extended periods (Bilbao et al., 2021, Doscher et al., 2022). Meccia et al., 2023 attributes these episodes to a multi-centennial oscillation triggered by the accumulation of salinity anomalies in the Arctic that, when released into the North Atlantic, affect water column stability and therefore convection. Importantly, they find that future scenarios with warmer climates lack sufficient sea ice to trigger the collapsing mechanism, potentially explaining its absence on* hist-piNTCF *members. Due to the strong dependency of the collapse episodes on internal variability, the specific response of convection to anthropogenic NTCFs is not correctly reflected in Fig. 4f and is likely underestimated.*"

12) Line 251: Could be worth noting here that this model only shows a decline after ~1980, at which point global aerosol concentrations had stabilized.

We thank the reviewer for this observation. The timing of the convection decline in MRI-ESM2-0 after ~1980 indeed aligns with the stabilisation of global aerosol

concentrations, which provides strong support for our central conclusion that NTCFs counteracted the GHG-driven convection decline.

The fact that MRI-ESM2-0 is the only model showing this clear temporal transition may reflect a higher sensitivity to GHG forcing. In fact, according to Bryden et al. 2024, MRI-ESM2-0 shows the strongest AMOC weakening (-67%) under SSP5-8.5 forcing (the ensemble includes EC-Earth3 and UKESM1-0-LL models, -34% and -50% respectively). However, other factors particular to each model, such as differences in the aerosol chemistry schemes or NTCF atmospheric lifetimes could be at play.

We think it is worth to include this information in the discussion, which now reads:
"*The* hist-piNTCF *experiments show a decrease in convection, in line with the expected response to rising GHG concentrations. In contrast, all* historical *experiments show stable or increasing* mlotst *values except for MRI-ESM2-0 (Fig. 5a). This model reports increasing convection until the 1980s after which convection declines, aligning with a first period of increasing global aerosol concentrations followed by a second period with stabilised aerosol concentrations and stronger GHG forcing. This suggests NTCFs counteracted, or at least mitigated, the GHG-driven decline in convection.*"

13) Fig. 6 caption: The variable name for salinity seems to have been entered incorrectly, i.e., (b, e) salinity (textitso).

Corrected.

14) Lines 262-263: Missing parenthesis, i.e., (as observed in Fig. 4).

Corrected.

15) Lines 265-269: This part seems too speculative to me. Can you present any evidence that this recirculation of saltier subsurface water is actually happening in the models? Or at least some citation from the literature supporting the existence of this positive feedback in the Labrador Sea?

A limitation of using free running coupled model simulations is that it is not possible to cleanly separate the driving signals of deep ocean convection from the ocean response to the associated vertical mixing. Therefore, our interpretation is unavoidably speculative to some degree.

Conceptually, NTCFs radiative forcing directly affects surface temperature whereas surface salinity anomalies can only emerge indirectly, e.g., via changes in the vertical mixing or the ocean circulation. We propose the presence of a convection-driven salinity feedback as an explanation of the maintained salinity increase, as such a mechanism has been identified in more idealised setups. In particular, Lenderink and Haarsma (1994) using both a one-box model and a conceptual ocean model, show that when convection is triggered in a region with cold and fresh surface waters above saltier and warmer subsurface waters, it produces a maintained convection state, with vertical mixing resulting in a saltier and warmer surface. Because the surface heat anomaly is rapidly lost to the atmosphere, while the salinity anomaly is conserved, the resulting positive density anomaly sustains convection. This feedback regime aligns with the properties of the Labrador Sea water column, and is consistent with the response to NTCFs seen in our results (Figs. 6 and B5, see also answer

to comment 17). However, given the limitations of our methodology and the lack of idealised experiments, we agree that the strength of our claim should be moderated in the revised text.

Lines 265-269 now read:
"*The saltier surface conditions may result from a positive feedback: stronger convection, initially driven by surface cooling, brings saltier subsurface waters to the surface, further increasing surface density and reinforcing deep convection. Although our analysis based on monthly model outputs does not allow us to clearly separate the driving signals of deep convection from the resulting response, a similar feedback mechanism has been identified in idealised frameworks (Lenderink and Haarsma, 1994), suggesting that this process is plausible in regions such as the Labrador Sea where subsurface waters are climatologically saltier (Fig. 6b). This mechanism could also explain the steady increase in* mlotst *seen in Fig. 5b, despite aerosol reductions after the 1980s.*"

16) Fig. B5 caption: Should be "temperature's contribution to density (sigmaT)".

Corrected.

17) Lines 270-275: I think that Fig. B5 is useful for understanding the contributions of temperature and salinity to the simulated density anomalies. However, I don't agree with the authors' interpretation of this figure. Specifically, it is stated that "temperature initially triggers surface density increases, which are subsequently reinforced by a salinity-driven feedback" (echoing a similar statement on lines 265-269). However, in October, temperature and salinity contribute about equally to the surface density anomalies. So, I don't see how Fig. B5 can be used to argue that temperature anomalies are the initial trigger of the density anomalies, and that salinity anomalies are a subsequent feedback. I think this section (and lines 265-269, which make similar statements) needs to be reworded a bit.

We thank the reviewer for this careful reading of Fig. B5, which led us to refine both our analysis and its interpretation. To provide a clearer view of the evolution of density anomalies, we modified Fig. B5 to include not only the convective months (October–May, in blue) but also the non-convective months (June–September, in red), following the climatological cycle shown in Fig. B3.

In the revised figure, the response of *sigmaT* to NTCFs peaks in August, far exceeding that of *sigmaS* (new Fig. B5e,f). This is consistent with the seasonality of irradiance and therefore with the seasonality of NTCF radiative forcing, which is strongest in summer in the Northern Hemisphere. During this season, warmer surface temperatures maintain water-column stratification. Therefore, stronger surface cooling due to NTCF forcing in the *historical* ensemble would increase surface density and erode the stratification, favouring convection in the following months. Once convection is active in October, the thermal contribution decreases while the haline contribution increases, consistent with enhanced vertical mixing bringing warmer and saltier subsurface waters to the surface.

[Figure]

**New Figure B5.** To enhance figure clarity only data from every second month is shown. Colours represent the state of climatological convection according to Figure B3: active (Oct-May; blue) and nonactive (Jun-Sep; red).

We also note the potential role of sea ice in shaping *sigmaS* signal. As shown in Fig. B2, NTCFs increase Labrador Sea sea ice extent in at least three models. Greater sea ice cover in the *historical* ensemble compared with *hist-piNTCF* would be expected to increase surface salinity through brine rejection during ice formation (autumn and winter), and reduce it through freshwater input when the ice melts (spring and summer). This seasonality qualitatively aligns with the evolution of the *sigmaS* response to NTCFs (new Fig. B5f). While a targeted analysis of brine injection or salinity transport would be required to quantify the sea ice role, we acknowledge its potential contribution. However, the relatively small seasonal variability of the haline contribution, as well as the larger magnitude of the summer thermal signal, suggest that the salinity response mainly reflects a feedback of intensified convection rather than a driving signal external to the convection system.

We therefore expanded the discussion in the revised manuscript to include the contributions to density during the non-convective months, and softened the strength of our statements to better reflect the limitations inherent to our analysis. Lines 270-275 now read:

"*The monthly evolution of potential density and its temperature and salinity contributions (Fig. B5; see subsection 2.3) provides additional insight into the processes driving convection, despite methodological limitations. We observe that NTCFs enhance the temperature contribution to surface density increase during the non-convective months (red profiles in Fig. B5e). During this summer period, warmer surface temperatures maintain water-column stratification, therefore, greater surface cooling due to NTCFs in the* historical *ensemble would increase surface density and erode the stratification, favouring convection in subsequent months. As convection activates, the relative contribution of salinity increases, consistent with enhanced vertical mixing bringing warmer and saltier subsurface waters to the surface. The seasonality of the* sigmaS *signal (Fig. B5f) is also consistent with the seasonal salinity changes arising from sea ice formation and melting, potentially relevant as the historical presence of NTCFs results in greater sea ice extent in the Labrador Sea region (Fig. B2). Further analysis would be required to quantify the sea ice contribution. However, the relatively small seasonal variability of the haline contribution, as well as the larger magnitude of the summer thermal signal, suggest that temperature anomalies are the dominant destabilising factor, while salinity anomalies reinforce and sustain convection.*"

18) Line 310: Should be "rsut", not "rust".

Corrected.

19) Fig. 9 and Fig. B6 captions: I'm a bit confused here about the distinction between MRI-ESM2-0 and the other models in terms of representing the effects of major volcanic eruptions. Even if the models other than MRI-ESM2-0 don't include interactive stratospheric chemistry, they should still prescribe the volcanic aerosols in their historical simulations, correct? If so, why isn't this reflected in od550aer? Do these models simply exclude the stratosphere in their calculation of od550aer? Or, is there some other explanation?

We agree with the reviewer that this distinction should be made more explicit. While some models explicitly resolve stratospheric chemistry, others parameterise the effects of volcanic aerosols. Unlike the other models, MRI-ESM2-0 includes stratospheric aerosols in the *od550aer* variable. As a result, peaks in AOD following major volcanic eruptions are present in both the *historical* and *hist-piNTCF* ensembles for the *od550aer* diagnostic (Fig. B6b). Despite this model-specific characteristic, the difference between these two experiments still effectively isolates the anthropogenic emissions signal, maintaining consistency with the other models in the study.

Evidence that all models in this study account for the radiative effects of volcanic aerosols is seen in Fig. B6a where all ensembles show a decrease in the *netR_HD* index after the major eruption of Mount Agung in 1963.

To clarify this point for readers, we have updated the figure captions:

-   Figure 9 caption: "*... Note that MRI-ESM2-0 (b) resolves stratospheric chemistry and therefore stratospheric aerosols are included in the* od550aer *variable. Regardless, all models in this study account for the radiative effects of volcanic aerosols either explicitly or through prescribed datasets or parameterisations.*"
-   Figure B6 caption: "*... Note that MRI-ESM2-0 resolves stratospheric chemistry and its effect is included in the* od550aer. *As a result, peaks following major volcanic eruptions are present. Regardless, all models in this study account for the radiative*

20) Lines 329-330: I would change this to "supports the hypothesis that aerosols, through some combination of direct effects and aerosol-cloud interactions, force…", or something similar. You haven't actually quantified the relative impacts of aerosol direct and indirect effects on the net radiation.

We agree that our original wording implied that our analysis distinguishes between aerosol direct and indirect effects, which it does not. Following the reviewer's suggestion, we have revised our discussion to reflect the combination of forcing pathways.

In addition, and following feedback from Referee #2, we expanded the analysis to include the NTCF signal on cloud radiative forcing (all-sky minus clear-sky fluxes), including the decomposition of shortwave and longwave radiation components. The results are shown in a new version of Figure 8 and discussed in the revised manuscript (see also our response to RC2 for further detail).

21) Lines 330-331: "Notably, the clt magnitude…" I don't understand this sentence. clt is the total cloud fraction/amount – how does it capture changes in other cloud properties besides that? And how can it be used to detect aerosol-cloud interactions? I would explain more clearly what you mean here, or just remove this sentence.

We thank the reviewer for raising this point. We acknowledge that our original statement was misleading: the total cloud fraction (*clt*) indeed represents only the percentage of sky covered by clouds and does not capture other relevant microphysical or radiative properties such as albedo or optical thickness. While *clt* could reflect changes in cloud lifetime, it is not an adequate or direct diagnostic of cloud properties. To avoid overinterpretation, we have removed this sentence from the revised manuscript.

22) Lines 335-339: I would remove this paragraph as it does not fit well within the rest of the discussion. First of all, you have not actually quantified aerosol-cloud interactions in your model simulations, so it's unclear how your results relate to those of Zhao and Suzuki (2021). Secondly, all of the previous discussion/analysis attempting to link aerosols to the ITCZ shift focused on the aerosol effect on the top-of-atmosphere radiation. Now, in this paragraph, you start to talk about aerosol effects on surface evaporation and the hemispheric atmospheric energy contrast. Again, I think this paragraph just doesn't fit well, adds confusion, and is unnecessary.

We agree that the paragraph in question does not align closely with our results and, as the reviewer notes, introduces unnecessary confusion. We have therefore removed this paragraph from the revised manuscript.

23) Lines 344-346: The Byrne et al. (2018) paper is a review paper that does discuss "not only changes in ITCZ location but also in its width and strength", in multiple contexts (e.g., observations, future climate projections). The role of aerosols is discussed some, but mainly (as far as I can tell) in terms of aerosol effects on ITCZ latitude. Please explain more clearly how your finding here of a negative correlation between netR_HD and equatorial rainfall amount is "consistent with" the Byrne et al. (2018) study. Or just remove this sentence.

After revisiting Byrne et al. (2018), we agree that the link we drew was not sufficiently supported in the context of our results. We have removed this sentence from the revised manuscript.

24) Lines 370-372: As discussed in previous comments, I don't believe that your results support these statements. The 38% increase in convection refers to the Feb.-Mar.-Apr. (FMA) season. During FMA, surface density (and thus convection) anomalies are driven primarily by salinity anomalies, not temperature anomalies (Fig. B5). And you've provided no evidence as far as I can tell to support the existence of the salinity feedback that is proposed here.

We agree that our conclusions regarding the salinity feedback mechanism, as currently phrased, may be too speculative given the evidence presented. The hypothesis is discussed in the context of our results as referenced in previous comments, however, we have removed it from the main conclusions of the paper. Lines 370-372 now read:

"*Increased Labrador Sea Convection: Over the period 1950–2014, historical NTCFs contributed to a 38% increase in FMA Labrador Sea convection. This increase is consistent with summer surface cooling eroding water-column stratification and favouring deeper convection in the following months. Once convection is active, salinity anomalies appear to lead the density response.*"

**Response to Referee #2**

The authors present an analysis of the CMIP6 AerChemMIP hist-piNTCF experiment, using an ensemble of 4 earth system models each with 3 ensemble members. By comparing against the CMIP6 hist experiment they explore the coupled climate response to historical aerosol and reactive gas emissions, focusing on the period from 1950 onwards when aerosol emissions rapidly increased, and on the impacts to global temperature patterns, Arctic sea ice, North Atlantic ocean convection, and the ITCZ.

It is great to see the piNTCF experiment being further utilised, and insofar as this paper documents the response of the latest generation of earth system models to historical near-term climate forcers in combination, it is a useful and overdue addition to the literature.

I have some suggestions for areas that the manuscript could be improved, as well as some comments of a more technical nature which are mostly seeking clarification or further detail. I would note that my expertise is mainly around atmospheric composition and particularly aerosol-climate interactions, and so I did not feel able to assess the part of the manuscript that discusses ocean circulation responses.

We thank Referee #2, Dr. Matthew Kasoar, for the time devoted to reviewing our manuscript and for the feedback provided. We greatly value the positive assessment of our contribution, as we agree that the AerChemMIP hist-piNTCF experiment offers an important opportunity to study the influence of anthropogenic aerosols and reactive gases on historical climate. In this document, we address each of the reviewer's comments and describe the corresponding revisions made to the manuscript.

My overarching comment on the manuscript is that while it adequately documents the climate response in the CMIP6 ensemble to historical NTCFs, which is valuable, the analysis mostly stops short of really attributing the responses, instead mainly relying on a review of existing literature and mechanisms that have been previously reported, to 'attribute' the responses to aerosol forcing, primarily on the basis that the response matches what would be expected and has previously been found for historical aerosol perturbations. In this respect, the results do not offer much new insight, although again in terms of documenting the CMIP6 response to piNTCF I still consider the contribution valuable.

It should be noted that the role of aerosols in driving the cooling trend between 1950-1980 in CMIP6 historical simulations has already been explored using AerChemMIP data by Zhang et al., 2021, (https://doi.org/10.5194/acp-21-18609-2021), and this paper ought be cited in appropriate places. The aerosol-driven cooling trend is one of the central results that the present study highlights, and the authors suggest that the observed pattern must be due to aerosols, but don't actually attribute this because they do not separate out the aerosols from reactive gas emissions. Comparison with the results of Zhang et al. based on the AerChemMIP hist-piAer experiment would therefore be a very useful comparison, and it's surprising that this study isn't referenced currently. Similarly the spatial pattern of historical aerosol cooling in CMIP6, with stronger cooling in northern high latitudes, is presented in AR6 WG1 Chapter 6 (Szopa et al. 2021) and this should probably also be cited.

Our study aims to assess the climate impact of combined NTCFs. With the used methodology, we are able to capture the interactive nature of these interconnected gas

phase and aerosol species. This combined perspective is of particular value from a policy standpoint, as it more faithfully represents the reality of co-emissions and therefore enhances the applicability of our results to decision-making. By quantifying these combined climate impacts, our analysis complements existing studies that have disentangled the responses to individual forcers, on which we rely to discuss attribution. In addition, while previous studies have mainly focused on global mean temperature, precipitation, and forcing, our work extends the analysis by quantifying the impacts of combined NTCFs on specific climate systems (i.e., the Arctic, the Labrador Sea, and the ITCZ). This regional focus adds further value to existing literature. We are therefore grateful for the reviewer's suggestion of Zhang et al. (2021) and the AR6 Chapter 6 (Szopa et al., 2021), which we now cite explicitly in our revised manuscript.

In particular, Zhang et al. (2021) use an analogous methodology by comparing the *historical* and *hist-piAer* experiments with an ensemble of ESMs overlapping with ours, making their study especially relevant to our discussion. They demonstrate that the anomalous global cooling reported in CMIP6 *historical* simulations between 1960 to 1990 is attributable to aerosol forcing. They also highlight the tendency of CMIP6 models to overestimate historical aerosol-induced cooling due to excessive sulfate loading, a caveat that we now mention in our analysis. Finally, they show that aerosol-cloud interactions (ACIs) account for most aerosol forcing sensitivity and are the main source of inter-model variability, providing useful context for our discussion of the radiative and cloud cover response to NTCFs (Section 3.4).

Additionally, Szopa et al. (2021) provide a broader assessment of NTCFs and their impacts. Their emission-based effective radiative forcing (ERF) analysis identifies $SO_2$ as the main source of net negative forcing among SLCFs (and to a lesser extent NOx and OC), consistent with our conclusion that aerosols are the primary drivers of the NTCF cooling signal. Regarding the spatial pattern, they show that aerosol concentrations since 1850 produce the strongest negative ERF over and downwind most industrialised regions in the Northern Hemisphere, resulting in a highly hemispheric signal. Similarly, the surface air temperature response to aerosols obtained from comparing *historical* and *hist-piAer* data shows a stronger cooling in the NH, peaking at polar latitudes. This Arctic amplification of the temperature response to aerosols closely resembles our findings for the NTCF cooling signal (Figs. 1b, 2b, 3, B1).

In the revised manuscript, we have moderated our attribution statements and supported our interpretation of the regional NTCFs responses through a discussion of other studies that disentangle the effects of individual forcings.

As a result, a related limitation is that the rationale for using hist-piNTCF in the present study isn't very clear. There is minimal discussion and no quantitative analysis of the climate impacts of tropospheric ozone and other reactive gases, and the authors conclude that the overall NTCF response is probably dominated by aerosol forcing. So the value added from using piNTCF, rather than piAer which has already been analysed in Zhang et al, is unclear. Using the combination of the two experiments (i.e. piNTCF compared with piAer) would have been nice to separate out the contributions of aerosols and ozone precursors to the overall NTCF response, but as it is the focus is entirely on aerosols and an assumption that these drive all the response.

As previously mentioned, the aim of our study is to document and quantify the overall impacts of NTCFs across a broad range of climate processes and systems. The *hist-piNTCF* experiment is well suited for this purpose, as it captures the non-linear interactions between species, which is particularly relevant for co-emitted forcers. This combined perspective is informative for decision-making, since it reflects the integrated impact of historical anthropogenic NTCFs, and can be regarded as an upper-bound or "extreme case" of their effects.

In our analysis, we focus on the similarity of the NTCF signal with the known aerosol forcing response and, as the reviewer notes, provide limited discussion of the role of tropospheric ozone and other reactive gases. We agree that a complementary analysis comparing *hist-piNTCF* and *hist-piAer* would be valuable to disentangle the aerosol contribution from the other species and it would be a great extension for future work. However, we did not pursue this here as it would have required reducing the ensemble size or breaking the consistency between the different sets of experiments, which would have compromised their comparability.

Our ensemble choice was guided by two requirements: the inclusion of models with interactive tropospheric chemistry and aerosols, and the availability of at least three ensemble members per model to filter out the uncertainty arising from internal variability. We adopted three realisations as the minimum recommended by the AerChemMIP initiative (Collins et al., 2017), although it has been argued that even larger ensembles would be preferable (Griffiths et al., 2025). Model selection was further refined by balancing ensemble size with the availability of variables of interest (e.g., chemistry, ocean and sea ice variables). Given all these limiting factors, we focused our analysis on *hist-piNTCF* data in order to maintain the largest possible ensemble and to ensure robust coverage of the Earth system responses we quantify.

We acknowledge, however, that the discussion should be complemented by existing attribution studies, and have accordingly expanded the discussion with additional references.

The authors loosely throughout the paper use phrases like 'we attribute this cooling to higher aerosol concentrations' (e.g. L204) - but, has it actually been attributed? 'Attribution' can have a specific meaning which is not really what the authors are doing here, I don't think. Although I completely agree this is the reason, nonetheless as currently presented this is really a hypothesis/assumption/interpretation, based on the fact that the cooling response is what we would expect to see, and what previous studies have shown, due to aerosols.

As noted in our previous responses, we have moderated these statements to ensure scientific accuracy and frame our interpretation within the context of existing attribution studies.

For example, in L230-231: "Overall, our results suggest that the primary driver of the Arctic response to NTCFs during 1950–1980 was high aerosol concentrations, with tropospheric ozone playing a secondary and opposing warming role" - I'm unclear how the results actually show this. The results show that the response to NTCFs (i.e. aerosols and ozone precursors combined) during 1950-1980 was a global cooling with a stronger response in the northern hemisphere. No results are shown which demonstrate that this coincided with high aerosol concentrations, or that tropospheric ozone contributed an opposing warming. While I

completely agree that this is certainly the case, and is probably expected background knowledge for any reader working in this field, nonetheless in the present study it's an inference drawn from other literature and the expected behaviour, not something that is actually demonstrated. Comparing the piNTCF with the piAer experiment and/or the results of Zhang et al. would be one way to ascertain this, and would enable the contribution of aerosols and ozone precursors to be separated, which would then be a much more holistic analysis of the piNTCF experiment. Showing the distribution of aerosols and tropospheric ozone, and how their emissions and/or burden has changed over time and how this corresponds (or doesn't) with the timeseries of temperature response in Figure 3, could be another way to help motivate such a conclusion.

Indeed, it seems an omission that no timeseries of the emissions is included, given that the authors discuss how the cooling trend is predominately during the 1950-1980 time period which corresponds to when aerosol emissions were increasing, before they then plateau, whereas ozone precursor emissions continue to increase. But these emission timeseries are never actually shown, so the reader is unable to make a judgement on whether the temperature trends really do align with the aerosol emission trends or not. E.g. L208-209: "This trend change is particularly evident in the differences between ensembles (Fig. 3b,d), which closely follow historical aerosol concentration trends" - it would be nice if this was shown (i.e. the historical aerosol concentration trends) so that the reader can see this for themselves, rather than just asserting it.

To complement the discussion and allow readers to directly evaluate the temporal alignment between NTCF concentrations and the simulated climate signals, we have added a new supplementary figure showing the time evolution of global-mean tropospheric ozone concentrations (at 500hPa) and aerosol optical depth at 550nm. Given their forcing relevance, these are the main quantities discussed throughout the manuscript.

The new Supplementary Figure illustrates that historical aerosol burdens increased between 1950 and 1980 and stabilised thereafter, whereas tropospheric ozone continued to rise over the same period. This temporal evolution is consistent with the surface air temperature (*tas*) response shown in Fig. 3b,d, where the difference between experiments grows during 1950–1980 and then weakens as aerosol forcing stabilises. In fact, a strong anticorrelation of -0.86 highlights the coupling between the *od550aer* and *tas* global signals across the ensemble members available for both variables (see new Sup.Fig. caption). The continued increase in ozone and other greenhouse gases (GHGs) provides a plausible explanation for the offset of the cooling trend after 1980. Additional evidence supporting the role of aerosols is that the "pothole-shaped" temperature evolution seen in historical experiments between 1960 and 1990 (Fig. 3a,c) has been attributed to aerosols by Zhang et al. (2021), as previously noted by the reviewer. We have clarified this link in the revised discussion.

For completeness, we note that the emissions for each experiment are prescribed following the CMIP6 AerChemMIP protocol (Collins et al., 2017), and the time evolution of different NTCF species emissions is documented in Hoesly et al. (2018). Furthermore, a regional decomposition of the emissions can be found in Szopa et al. (2021, Fig. 6.19).

[Figure]

[Figure]

**New Supplementary Figure.** Annual global mean evolution of the (a) ozone concentration (*o3*) at 500hPa and (b) aerosol optical depth at 550nm (*od550aer*) for the *historical* (solid lines) and *hist-piNTCF* (dashed lines) CMIP6 simulations over the period 1950–2014. Colours represent data from individual models (BCC-ESM1: blue, EC-Earth3-AerChem: green, MRI-ESM2-0: yellow, UKESM1-0-LL: pink). Each model mean is obtained from 3 different members but for UKESM1-0-LL od550aer data, which only has 2 members available, and BCC-ESM1 with no data available for this variable. Note that MRI-ESM2-0 resolves stratospheric chemistry and its effect is included in the *od550aer* diagnostic. As a result, peaks following major volcanic eruptions are present. Regardless, all models in this study account for the radiative effects of volcanic aerosols either explicitly or through prescribed datasets or parameterisations.

Additional minor comments:

- As well as the Collins et al. (2017) AerChemMIP protocol paper, the recent AerChemMIP retrospective which summarises the outcomes of the initiative, by Griffiths, Wilcox, Allen et al., 2025, (https://doi.org/10.5194/acp-25-8289-2025) should probably also be cited in the introduction when introducing AerChemMIP.

We agree with the reviewer's suggestion. The citation to Griffiths et al. (2025) has been included in the introduction as follows:
"*The Aerosols and Chemistry Model Intercomparison Project (AerChemMIP; Collins et al., 2017; Griffiths et al. 2025), endorsed by CMIP6, specifically targets NTCFs to quantify the climate and air quality impacts of aerosols and chemically reactive gases through a range of dedicated simulations.*"

- L39: "attributed to meridional forcing gradients in midlatitudes" - is it the gradient in the midlatitudes that matters, or the interhemispheric difference?

Revisiting Allen and Sherwood (2011), we agree the wording should be clarified when explaining their findings on the ITCZ response to aerosol forcing. More specifically, they identify interhemispheric meridional temperature gradients driven by heterogeneous radiative forcing as the underlying mechanism for the circulation shift, both for scattering and absorbing aerosols.

Lines 35–39 now read:
"*Allen and Sherwood (2011), through sensitivity experiments with an atmospheric general circulation model, found nearly opposite responses in atmospheric circulation to radiation scattering or absorbing aerosols. While scattering aerosols reduce the Hadley cell width and displace the Intertropical Convergence Zone (ITCZ) southward, absorbing aerosols lead to a*

*northward ITCZ shift. These responses are attributed in their study to interhemispheric temperature gradients arising from the spatially uneven distribution of the radiative forcing.*"

- Figures with maps, e.g. Figure 1, Figure B8 - please include global mean values for reference as well

We thank the reviewer for this suggestion. We have updated the figures to include global mean values (new Figs. 1, B8). In addition, new Figure B2 now shows the change in sea-ice extent in the *historical* experiment relative to the *hist-piNTCF* experiment.

[Figure]

**New Figure 1.** Global mean values for each magnitude are shown in gray in the title.

[Figure]

**New Figure B2.** Values of sea ice extent change (ΔSIE, shown in gray above each panel) denote the percentage difference in total area with siconc ≥ 15% (sea ice extent) between *historical* and *hist-piNTCF* experiments, relative to the *hist-piNTCF* extent.

- L215: "This aligns with previous studies (Wu et al., 2024)" - this is only one study, not multiple studies

Corrected

- L221-225: "Indeed, the observed cooling in Fig. 1b aligns with an increase in sea ice concentration. Examining boreal autumn data—when sea ice retreat peaks reveals a consistent increase in sea ice extent in the historical ensemble relative to hist-piNTCF across multiple models (Fig. B2). The strongest sea ice expansion in the Barents Sea corresponds to the most pronounced temperature decreases (Fig.1b), reinforcing the role of sea ice-albedo feedback in amplifying Arctic cooling" - but, an expansion of the Arctic sea ice would also be an expected consequence of Arctic cooling, so on it's own I'm not sure this provides much evidence one way or the other for a dominant role of the sea ice feedback mechanism as the main factor amplifying the cooling. Even if other factors are actually more important for driving the amplification, you would still expect to see a sea ice response, and so the presence of such a sea ice expansion neither proves nor disproves the hypothesis that it plays the dominant role.

We appreciate this observation and agree that an expansion of Arctic sea ice would indeed be an expected response to regional surface cooling, and its presence alone does not allow us to identify the dominant feedback mechanism responsible for amplifying the cooling signal. Several processes may contribute to this cooling and associated sea ice increase, including radiative changes, atmospheric and oceanic forcing, as well as feedback mechanisms previously linked to Arctic amplification. In particular, the Barents Sea—where the strongest signal is found—is known to exhibit high sensitivity to external forcing and large internal variability (Rieke et al., 2023, Siew et al., 2024), which may contribute to the pronounced local response.

However, our objective is to assess the effect of NTCFs on the Arctic climate system as simulated by the models, rather than to disentangle the relative contributions of these processes. Therefore, we agree that causal statements should be avoided, and now limit

ourselves to documenting the coinciding patterns of enhanced sea ice extent and surface temperature decrease, while acknowledging the combination of processes—including, but not limited to, the sea ice-albedo feedback—that govern Arctic Amplification.

Lines 221-225 now read:
"*Indeed, the observed cooling in Fig. 1b aligns with an increase in sea ice concentration (*siconc*). Examining boreal autumn data—when sea ice retreat peaks (Deser et al., 2010)—reveals a consistent increase in sea ice extent in the* historical *ensemble relative to* hist-piNTCF *across multiple models (Fig. B2). The strongest increase occurs in the Barents Sea, a region known for its high sensitivity to external forcing and large sea ice internal variability (Rieke et al., 2023, Siew et al., 2024), and spatially aligns with the most pronounced temperature decreases (Fig. 1b). While this co-variability is consistent with the operation of local positive sea ice-related feedback mechanisms, the observed changes likely reflect the combined influence of several processes governing the cooling and its amplification (Previdi et al. 2021).*"

- L229: "A reverse mechanism — enhanced Arctic sea ice extent (Fig. B2) — could explain the observed tropical cooling" - enhanced tropical upper tropospheric warming is a well established temperature response to global warming, due to the lapse rate feedback. Vice versa, an enhanced cooling signal in the tropical upper troposphere will be the expected response to a global dimming caused e.g. by aerosols. Although Arctic sea ice expansion will also contribute as an additional feedback which adds to the global cooling and the tropical signature, I don't see any reason to think this is the main reason rather than the well-established lapse rate feedback pattern that would accompany any global cooling.

We appreciate this important clarification. As the reviewer points out, the tropical atmosphere is characterised by moist convection and associated lapse-rate adjustments that amplify surface temperature changes in the upper troposphere (Colman and Soden, 2021). In our case, surface cooling due to NTCFs (Fig. 1b) would reduce water vapour content in the atmospheric column and steepen the vertical temperature gradient, producing a stronger cooling signal in the tropical upper troposphere (Fig. 2b). The lapse rate feedback refers to the stabilising effect of this vertical structure, as colder upper levels emit longwave radiation less efficiently, thereby limiting the overall tropical cooling.

We agree with the reviewer's concerns and now explain the temperature signal as a result of lapse rate adjustments inherent to the tropical atmosphere. At the same time, additional processes may affect tropical temperature beyond the direct cooling from NTCFs. As cited in the original manuscript, England et al. (2020) link Arctic sea ice loss to reduced subtropical ocean circulation, weakened equatorial upwelling, and tropical warming. Analogously, an Arctic sea ice increase could reinforce the tropical cooling we detect. Similarly, Sun et al. (2018) isolate the effects of Arctic sea ice loss under the RCP8.5 scenario and find that Arctic sea ice reductions induce a "mini-global warming", highlighting the role of sea ice as an amplifier of global temperature changes. Furthermore, Screen et al. (2018) use different coupled ocean-atmosphere models and perturbation protocols to show Arctic sea ice loss affects tropical temperatures through change in ocean heat transport, creating sea surface anomalies that are then magnified at higher levels.

All in all, as discussed in previous responses, our analysis does not allow us to quantitatively disentangle the contribution of individual climate processes, and our interpretations should therefore be understood as a qualitative description of the impacts rather than attribution. In the revised text, we now provide broader context on tropical temperature processes, emphasising the likely primary role of lapse rate adjustments in shaping the NTCF-induced tropical signal, and framing the Arctic sea ice expansion as a possible secondary contributor.

Lines 226-229 now read:
"*Additionally, Arctic sea ice expansion may contribute to the tropical cooling signal observed in Fig. 2. This vertical structure, showing amplified temperature anomalies in the tropical upper troposphere, is consistent with the operation of moist-adiabatic lapse rate adjustments (Colman and Soden, 2021). The pattern also aligns with the response to Arctic sea ice loss reported by England et al. (2020). They link Arctic sea ice loss to a slowdown in subtropical meridional ocean circulation, reducing equatorial upwelling and warming the tropical atmosphere, suggesting that the enhanced sea ice extent in the* historical *ensemble (Fig. B2) may further contribute to the observed cooling.*"

- L232-234: "The amplified cooling in the Arctic is largely mediated by sea ice feedbacks, though additional factors, such as regional radiative changes, as well as, variations in atmospheric, and oceanic energy transport, may also contribute to the observed temperature changes" - again, this is speculative based on mechanisms that have been reported in previous studies, rather than something which is demonstrated by any of the results presented here. There are multiple mechanisms that are known to contribute to Arctic Amplification, and based on the results shown here, the authors have no basis for concluding that sea ice feedbacks are the dominant driver, as far as I can see. I'm happy to be corrected if I've missed something which allows the contribution of different feedbacks to the overall Arctic cooling to be determined though.

We agree that our previous wording was misleading. We now explicitly state that, while our results are consistent with Arctic amplification, our study was not designed to quantify the relative role of the involved processes, and that such attribution would require targeted sensitivity experiments (see response to the previous comment referring to lines 221–225).

Lines 230-235 now read:
"*Overall, our results show that NTCFs produced Arctic cooling with strong amplification between 1950 and 1980. The observed increase in sea ice extent spatially aligns with the temperature response, which could suggest the operation of sea ice–related feedback mechanisms. Other processes that have been previously invoked to explain Arctic Amplification, such as changes in atmospheric and oceanic poleward energy transports (Iwi et al., 2012; Robson et al., 2022; Needham and Randall, 2023), cloud and water vapour feedback (Goosse et al., 2018), and lapse-rate feedback (Pithan and Mauritsen, 2014) may have contributed to the pronounced regional temperature changes. Quantifying their relative influence, however, lies beyond the scope of this analysis.*"

- L330-331: "Notably, the clt magnitude captures both changes in cloud amount and cloud properties, allowing for detection of potential aerosol–cloud interactions" - does it? Surely clt (cloud area fraction) is just the 2D cloud amount, i.e. the fraction of the gridbox covered by cloud. It doesn't tell you about other cloud properties that are affected (possibly even more

strongly so) by aerosol microphysics such as cloud albedo. If the authors were to calculate e.g. the change in cloud radiative forcing (difference between all-sky and clear-sky radiative fluxes), then this would capture both changes in cloud amount, and other properties like cloud albedo and optical thickness.

We thank the reviewer for raising this point. We agree that our original wording was inaccurate, and we have removed this sentence from the revised manuscript.

Due to the importance of aerosol–cloud interactions in the total aerosol radiative forcing (Szopa et al. 2021), we consider relevant to further explore the role of clouds in the detected NTCF signal. As suggested by the reviewer, we calculated the NTCFs signal for the all-sky and clear-sky radiative fluxes as well as for the cloud radiative forcing (Sup.Fig. 2). In the clear-sky total radiation signal (Sup.Fig. 2b), the negative anomalies in the Northern Hemisphere are restricted to continental areas with high anthropogenic aerosol emissions– namely North America, central Europe and East Asia. The presence of clouds, however, determines the zonal distribution of the radiative signals we see for the all-sky variable (Sup.Fig. 2a,c and Fig. B8 of the manuscript). From this additional analysis we can qualitatively conclude that NTCFs effect on clouds plays a critical role in shaping the interhemispheric radiative signal.

[Figure]

**Supporting Figure 2.** Impact of historical NTCF forcings on radiation fluxes at the top of the atmosphere (TOA). In particular: (a,b,c) net total, (d,e,f) net shortwave and (g,h,i) net longwave radiation (positive values for entering energy and viceversa). The panels show the difference in climatologies between the multi-model *historical* and *hist-piNTCF* ensembles over the period 1950–2014 for all-sky radiative fluxes (a,d,g), their clear-sky counterparts (b,d,f) and cloud forcing obtained by subtracting the clear-sky from the all-sky fluxes (c,f,i). Both ensembles are comprised of four models (BCC-ESM1, MRI-ESM2-0, UKESM1-0-LL, and EC-Earth3-AerChem), with three members each. Stippling is applied to significant values according to a two related samples t-test with a 95% confidence.

In the revised manuscript, we modified Figure B8 to include NTCFs signal on cloud forcing and added the insights derived from the additional analysis as follows:

"*To further contextualise the aerosol–radiation connection, Figure B8 shows a spatial alignment between net radiation and cloud forcing (difference between all-sky and clear-sky net radiation), weaker over polar regions, highlighting the relevance of clouds in shaping the radiative response to NTCFs. The global mean cloud radiative forcing (−0.59 W m⁻²) exceeds the net radiation signal (−0.42 W m⁻²), which is consistent with aerosol indirect negative forcing being partially compensated by the positive forcing of absorbing aerosols and tropospheric ozone. Although our analysis does not isolate individual forcing pathways this interpretation agrees with estimates of NTCFs effective radiative forcing (ERF) in previous studies. Thornhill et al. (2021) analysing AerChemMIP output found that NTCF ERF (−0.89 ± 0.20 W m⁻²) is dominated by aerosols (−1.01 ± 0.25 W m⁻²) over tropospheric ozone (0.03 ± 0.01 W m⁻²) and its precursors (0.20 ± 0.07 W m⁻²). Also using CMIP6 output, Forster et al. (2021) gave an estimation of aerosol ERF (−1.11 ± 0.38 W m⁻²) and found a prevalence of the aerosol-cloud interaction component (−0.86 ± 0.57 W m⁻²) with respect to aerosol-radiation interactions (−0.25 ± 0.40 W m⁻²). Aerosol-cloud interactions therefore are a dominant contributor to NTCF forcing, but also constitute a major source of inter-model uncertainty (Szopa et al., 2021; Zhang et al., 2021). While ERF values are not directly comparable to our coupled-simulation signals, they provide useful context and share the hypothesis that clouds play a central role in the radiative forcing of NTCFs.*

*Changes in total cloud cover (clt) also broadly align with the net radiation signal, particularly over mid-to high latitudes (Fig. B8c). The weaker spatial correspondence compared to that of cloud forcing (Fig. B8b) suggests that NTCFs affect not only cloud amount but also cloud properties. On the global scale, we obtain a mean increase in clt of +0.5% due to historical NTCFs, comparable to the values reported by Zang et al. (2021) for aerosol forcing. Consistent with Andersen et al. (2023), we find an inverse relationship between mid-latitude clouds and absorbed radiation due to the dominance of short-wave (SW) reflection (white stippling), while in tropical regions cloud long-wave (LW) retention prevails (black stippling). Additional SW–LW decomposition of the radiative fluxes (not shown) supports this interpretation. In the Northern Hemisphere, regions with high anthropogenic NTCF emissions—such as North America, Europe, and East Asia—and their downwind regions exhibit both increased cloud cover and, more strongly, negative cloud radiative forcing dominated by the SW component. This, combined with more localised clear-sky SW reflectivity increases, produces a predominantly zonal negative signal. In the Southern Hemisphere, clear-sky fluxes reveal a dominant role of the LW component resulting in a positive hemispheric signal, reinforced by cloud forcing: in the tropics, the increase in cloud cover leads to higher LW retention, whereas the decrease in mid-latitude cloud cover likely reduces SW reflectivity. Altogether, these results underline that cloud radiative forcing is a key contributor to the interhemispheric radiation imbalance previously identified.*"

[Figure]

**New Figure B8.** The impact of anthropogenic NTCFs on cloud forcing (all-sky minus clear-sky net radiation) has been added. Global mean values for each magnitude are shown in gray in the title.

- Please double check figure numbering - for example, it looks like fig. B8 is referenced in the text before figure B7 (though apologies if I've missed an earlier instance of it).

We thank the reviewer for spotting this error. With the addition of new figures in the revised manuscript, the numbering has been updated to ensure consistency with the order of references in the text.

- L341-343: "The analysis reveals a clear negative correlation, with higher netR_HD values associated with reduced equatorial rainfall. This suggests that an increased radiation imbalance between hemispheres, previously linked to a southward shift in the meridional circulation, also leads to a reduction in precipitation near the equator" - does it suggest this? Couldn't the reduced precipitation just be a consequence of globally cooler temperatures, rather than because of the hemispheric difference? Because the global cooling is caused by aerosols, which are preferentially located in the northern hemisphere, the hemispheric difference in netR will also be correlated with global temperature. So, this analysis can't distinguish between an effect caused by the hemispheric temperature difference, and an effect caused by globally cooler temperatures. The precipitation reduction could therefore just be due to the overall temperature reduction and not the interhemispheric difference; you will find a correlation either way. If the authors wish to argue that it is indeed the *gradient* of radiative forcing and not just the overall magnitude that drives reduced ITCZ precipitation, this needs to be backed up with additional analysis elucidating the mechanism for this, I think.

We agree with the reviewer that the current analysis does not allow us to distinguish between the effects of interhemispheric radiative imbalance and those of overall global cooling, and that our wording therefore overstated the causal link. As the reviewer notes, because aerosol's net negative radiative forcing is preferentially located in the Northern Hemisphere, netR_HD is inherently correlated with global temperature decrease, so reduced equatorial precipitation could result from either (or both) mechanisms.

To clarify this, we have performed an additional analysis comparing ITCZ precipitation with global mean temperature values. Precipitation is expected to scale approximately linearly with global temperature (Held and Soden, 2006) and while the results show this, the relationship is weaker than with netR_HD, especially in the later decades where temperature continues increasing but precipitation does not follow proportionally (Sup.Fig. 3).

Our interpretation is that the tropical precipitation response to NTCFs likely combines the effects of global cooling and atmospheric circulation changes related to the interhemispheric radiation imbalance. As documented by Allen and Sherwood (2011), such an energy imbalance can weaken the Hadley cell, which could contribute to the precipitation reduction. Additionally, the southward displacement of the ITCZ we identified may influence rainfall amounts by shifting the main convection zones between oceanic and continental regions. Fully disentangling the role of these factors would be an interesting direction for future work but beyond the scope of the present study.

We have incorporated these insights into the revised discussion as follows:
"*The analysis reveals a clear negative correlation, with higher* netR_HD *values associated with reduced tropical rainfall. Because NTCFs exert negative radiative forcing (Fig. B8a) and subsequently induce surface cooling (Fig. 1b) predominantly in the Northern Hemisphere,* netR_HD *is inherently correlated with the global temperature decrease. Therefore, Figure B7 could be reflecting the expected reduction in precipitation that accompanies the global cooling (Held and Soden, 2006). Additional analysis confirms this temperature-precipitation link, but we find it to be weaker than the relationship between precipitation and* netR_HD *(not shown). While the physical interpretation of this relation is complex, our results suggest that the tropical precipitation response to NTCFs likely combines the effects of global cooling and atmospheric circulation changes driven by the interhemispheric radiation imbalance. As shown by Allen and Sherwood (2011), such an imbalance can weaken the Hadley cell, which could contribute to the precipitation reduction. Additionally, the southward displacement of the ITCZ (Fig. 7b) may further modulate rainfall by shifting convection zones between oceanic and continental regions.*"

[Figure]

**Supporting Figure 3.** Relationship between the global mean surface air temperature (*tas*) and the ITCZ precipitation amount index. Analogous to Figures 8 and B7 of the original manuscript.